

# Corrosion-influencing microorganisms in petroliferous regions on a global scale: systematic review, analysis, and scientific synthesis of 16S amplicon metagenomic studies

Joyce Dutra[1], Rosimeire Gomes[1], Glen Jasper Yupanqui García[2], Danitza Xiomara Romero-Cale[3], Mariana Santos Cardoso[2], Vinicius Waldow[4], Claudia Groposo[5], Rubens N. Akamine[4], Maira Sousa[4], Henrique Figueiredo[6], Vasco Azevedo[7] and Aristóteles Góes-Neto[7]

[1] Graduate Program in Microbiology, Federal University of Minas Gerais, Belo Horizonte, Minas Gerais, Brazil
[2] Graduate Program in Bioinformatics, Federal University of Minas Gerais, Belo Horizonte, Minas Gerais, Brazil
[3] Graduate Program in Biotechnology, State University of Feira de Santana, Feira de Santana, Bahia, Brazil
[4] Petrobras Research and Development Center (CENPES), Petrobras, Rio de Janeiro, Rio de Janeiro, Brazil
[5] Petrobras, Criciúma, Santa Catarina, Brazil
[6] Veterinary School, Federal University of Minas Gerais, Belo Horizonte, Minas Gerais, Brazil
[7] Institute of Biological Sciences, Federal University of Minas Gerais, Belo Horizonte, Minas Gerais, Brazil

Corresponding author
Aristóteles Góes-Neto, arigoesneto@gmail.com

## ABSTRACT

The objective of the current systematic review was to evaluate the taxonomic composition and relative abundance of bacteria and archaea associated with the microbiologically influenced corrosion (MIC), and the prediction of their metabolic functions in different sample types from oil production and transport structures worldwide. To accomplish this goal, a total of 552 published studies on the diversity of microbial communities using 16S amplicon metagenomics in oil and gas industry facilities indexed in Scopus, Web of Science, PubMed and OnePetro databases were analyzed on 10th May 2021. The selection of articles was performed following the Preferred Reporting Items for Systematic Reviews and Meta-Analyses (PRISMA) guidelines. Only studies that performed amplicon metagenomics to obtain the microbial composition of samples from oil fields were included. Studies that evaluated oil refineries, carried out amplicon metagenomics directly from cultures, and those that used DGGE analysis were removed. Data were thoroughly investigated using multivariate statistics by ordination analysis, bivariate statistics by correlation, and microorganisms' shareability and uniqueness analysis. Additionally, the full deposited databases of 16S rDNA sequences were obtained to perform functional prediction. A total of 69 eligible articles was included for data analysis. The results showed that the sulfidogenic, methanogenic, acid-producing, and nitrate-reducing functional groups were the most expressive, all of which can be directly involved in MIC processes. There were significant positive correlations between microorganisms

in the injection water (IW), produced water (PW), and solid deposits (SD) samples, and negative correlations in the PW and SD samples. Only the PW and SD samples displayed genera common to all petroliferous regions, *Desulfotomaculum* and *Thermovirga* (PW), and *Marinobacter* (SD). There was an inferred high microbial activity in the oil fields, with the highest abundances of (i) cofactor, (ii) carrier, and (iii) vitamin biosynthesis, associated with survival metabolism. Additionally, there was the presence of secondary metabolic pathways and defense mechanisms in extreme conditions. Competitive or inhibitory relationships and metabolic patterns were influenced by the physicochemical characteristics of the environments (mainly sulfate concentration) and by human interference (application of biocides and nutrients). Our worldwide baseline study of microbial communities associated with environments of the oil and gas industry will greatly facilitate the establishment of standardized approaches to control MIC.

## INTRODUCTION

Microorganisms are present in virtually all oil and gas industry environments containing or interacting with free water. These environments range from the network of pores inside the rocks of oil reservoirs, to the surface and subsurface facilities and equipment that constitute the infrastructure of the oil and gas industry, such as injection wells, production wells, pipelines, valves, membranes, tanks, pumps, separators, water and effluent treatment units, and cooling towers (*Ollivier & Magot, 2005*; *de Waldow, 2018*). The microbial activities that occur in oil and gas industry environments can generate benefits (*e.g.*, bioremediation) or harms (*e.g.*, microbiologically influenced corrosion or MIC), depending on the microbial species and the prevailing environmental conditions (*Augustinovic et al., 2012*).

The global annual costs of the corrosion of pipelines and other industrial equipment for oil and gas production and transport have been estimated at 2.5 trillion US dollars per year, according to the *NACE International (2016)*, and it is estimated that between 10% and 20% of corrosion occurrences involve MIC (*Machuca & Salgar-Chaparro, 2019*). MIC refers to corrosion that is affected by the presence and activity of microorganisms, which involves the establishment of biofilms that initiate and/or accelerate corrosive processes of electrochemical nature (*Eckert, 2014*; *Videla, 1996*; *Zarasvand & Rai, 2014*). MIC is difficult to predict and monitor due to the heterogeneity of biofilms distribution within industrial systems and because it induces localized corrosion, whose monitoring is much more challenging than general corrosion (*Little & Lee, 2007*; *Roche, 2007*). Operating expenditures for mitigating MIC include mechanical cleaning, pigging, injection of chemical products (*e.g.*, biocides, biodispersants, corrosion inhibitors, $O_2$ and $H_2S$ scavengers), line inspection, and microbiological and physicochemical analyses (*Skovhus, Eckert & Rodrigues, 2017*).

Microbiological analyses allow the evaluation of the microbiota present in a given environment, including those that are capable of causing or accelerating corrosion. Although cultivation-dependent methods are still the most widely used in the oil and gas industry, they are limited because cultivable lineages usually do not exceed 1.0% of all microbial lineages present in environmental samples (*Pedrós-Alió, 2011*). In contrast, molecular microbiological methods (MMM) are culture-independent methods that have attracted the interest of the oil and gas industry for monitoring microbial communities. These methods are based on biomolecules such as adenosine triphosphate (ATP), deoxyribonucleic acid (DNA), ribonucleic acid (RNA), proteins, and metabolites (*Skovhus, Eckert & Rodrigues, 2017*). The advent of MMM in the oil and gas industry is making it possible to obtain a more comprehensive and accurate picture of the microbial communities occurring in different environments related to the oil and gas industry (*Larsen et al., 2009*).

The term "metagenomics" (first coined and used in 1998) refers to the sequencing of DNA molecules extracted directly from environmental samples, allowing a genomic investigation of microbial communities and bypassing the need of cultivation in the laboratory (*Handelsman et al., 1998*). For many years since its first experimental use, metagenomic analysis has relied on extracting community DNA from a sample, using restriction enzymes to cleave the DNA molecules, followed by cloning DNA segments into vectors and scanning the inserts in thousands of clones from metagenomic libraries to explore new products and biological activities (*Handelsman et al., 1998*). The development of the so-called Next-Generation Sequencing (NGS) technologies, combined with improvements in software for bioinformatics analyses, allowed an immediate increase in the number of sequenced metagenomes and at lower costs, contributing to the dissemination and improvement of metagenomics (*Scholz, Lo & Chain, 2012*; *Stark, Giersch & Wünschiers, 2014*).

One limitation of metagenomics through analysis of 16S rRNA gene amplicons (also known as metabarcoding) is the absence of information about the functional and metabolic capabilities of the evaluated communities (*Douglas et al., 2020*). The prediction of metabolic functions can assist in the interpretation of metagenomic analysis of various environments and their respective microbial communities (*Ijoma et al., 2021*). One tool that has been used is software that can predict metabolic and functional profiles of microbial lineages based on taxonomic identification through 16S rRNA sequences and retrieval of published studies on metabolic characterization of microbial lineages. An example of this kind of software is PICRUST2, that includes a vast and updated database of gene families and reference genomes, allowing phenotypic predictions from amplicon metagenomes (*Douglas et al., 2020*). In complement to metagenomic analyses, physicochemical analyses allow us to understand which factors trigger the proliferation or demise of microbial lineages in a specific environment (*Gao et al., 2019*).

Nevertheless, although those tools provide us with information that can predict and diagnose MIC problems, it is still far to thoroughly understand all the intricate interactions comprising MIC. Literature data are heterogeneous, and it is often not possible to find a common denominator that can explain the putative driving factors that may be influencing

the community dynamics of distinct microorganisms in each environment. In addition, obtaining data related to microbial communities present in industrial infrastructure is limited due to logistic difficulties of access to facilities and sample collection in pipelines, submarine, and subsurface equipment (*Rachel & Gieg, 2020*). Furthermore, the lack of communication between different departments within the same company, particularly in large companies (*Kunsch, 2012*), hinders the access to information by scientists and engineers.

Given that scenario, it is highly relevant to establish a standardized study methodology, so that it is possible to compile and analyze the bulk of information already published, since there are still many knowledge gaps regarding the microbial communities' structure and dynamics in oil and gas infrastructure. Therefore, the present systematic review aimed to evaluate the composition and relative abundance of bacteria and archaea associated with MIC, as well as the prediction of metabolic functions present in different samples from oil production and transport systems worldwide.

## METHODS

### Data collection

This systematic review was based on the Preferred Reporting Items for Systematic Reviews and Meta-Analyses (PRISMA) guidelines (*Moher et al., 2009*). All records from all types of documents, of all languages, available as digital media since the beginning of Scopus, Web of Science (main collection), PubMed and OnePetro (focused on the oil and gas industry) databases up to 10th May 2021, were retrieved. The study selection process was performed in three steps: (1) identification of records in Scopus, Web of Science, PubMed and OnePetro databases; (2) screening of documents; and (3) evaluation of eligibility and inclusion of selected studies.

In order to identify the records, a text search evaluating all the documents in all the three databases was carried out, using adequate and standardized search patterns. The keywords searched for were: petroleum, crude oil, oilfield, oil, reservoir, pipeline, biocorrosion, microbiologically influenced corrosion, microbial corrosion, MIC, corrosion, metagenomics, and 16s rRNA. All keyword combinations and the Boolean operators are detailed in Supplementary Material 1.

Identified documents were automatically exported from Scopus, Web of Science and PubMed in *csv* text file format, whereas those from OnePetro were manually exported (Supplementary Material 2). Using the script *format_input.py* (Supplementary Material 3), Excel spreadsheet files were generated for each of the three databases, including the following variables: Title, Year, Digital Object Identifier (DOI), Document Type, Language, and Author(s). The script identified records without DOI and removed those ones with identical Title or DOI.

Excel spreadsheet files of each database were unified, and duplicated records were removed using the script *remove_duplicates.py* (Supplementary Material 4). Besides the initial variables, the unified spreadsheet file was added with the variable Repository, which describes from which database the record was retrieved. Furthermore, when the same record was identified in more than one database, this information was maintained in the

 

**Table 1 Eligibility criteria for inclusion of studies in the systematic review.**

| Eligibility criteria | |
|---|---|
| Collection point | Reservoirs, pipelines, and tanks |
| Sample | Oil, produced water, injection water, biofilm, sediment, sludge and pig residue |
| Result | Composition and abundance of microorganisms present in oil field samples by 16S amplicon metagenomic analysis |
| **Exclusion criteria** | |
| Collection point | Refinery and petrochemical industry |
| Sample | Obtained from cultures |
| Result | Composition and abundance of microorganisms present in oil field samples by molecular analysis of the Denaturing type Gradient Gel Electrophoresis (DGGE). |

description. The documents were downloaded as *pdf* files by their DOIs using a software based on permissions of ICB/UFMG internal network. Inaccessible documents were categorized as *Not Available (NA)* whereas the documents without DOI were manually downloaded.

The selection of studies was performed by Joyce Dutra (J.D.) and Rosimeire Gomes (R. G.) and the divergences were solved by Mariana Santos Cardoso (M.S.C.). The divergences were solved as follows: After obtaining the documents (J.D. and R.G.), selected and non-selected files were merged, and the discrepancies were identified. Thus, these divergent documents were separated in one spreadsheet and forwarded to the third reviewer (M.S.C.), who judged whether the document would enter or not. In case of doubt, there was a census in the group of authors to decide whether the document should be remained in the final *corpus* or removed.

Titles and abstracts of the documents were reviewed, and only studies related to both oil fields and metagenomic analyses were retrieved. All these studies were completely re-evaluated based on the eligibility and exclusion criteria related to collection area, sample type, and obtained results: inclusion and exclusion criteria are listed in Table 1.

The following data were extracted from selected documents: (i) sampling design, (ii) physicochemical characterization of samples, (iii) DNA extraction, amplification, and high-throughput sequencing features, and (iv) taxonomic composition (all taxa from Domain to Genera) and relative abundance of microorganisms obtained by both 16S rRNA amplicon and shotgun metagenomics.

## Data analyses
### *Overview*
In order to standardize the data, all analyses encompassing the relative abundance of microbial taxa in all samples were performed at the genus level (even if there were documents including classification at lower-level taxonomic ranks, such as the species level).

For data extraction, all archaeal and bacterial genera were considered, regardless of their originally described relative abundance or absence of this information (in case there was only the report of their presence). After data extraction, absolute and relative frequencies for each genus were calculated. Data analyses were carried out for each sample type separately (injection water—IW, produced water—PW, oil— OIL, oil and water—OW, and solid deposits—SD, which include biofilms, sludge, and pigs). Each set of sample types was evaluated for each distinct world petroliferous region, following the United States Geological Survey classification (*USGS, 2003*): 1—Former Soviet Union, 2—Middle East and North Africa, 3—Asia-Pacific, 4—Europe, 5—North America, 6—Central and South America, 7—Sub-Saharan Africa and Antarctica, and 8—South Asia.

### Shareability and uniqueness patterns

Results of analyses of shareability and uniqueness patterns were illustrated as Venn diagrams, which were built with Venn v1.10 R package (*Dusa, 2017*). In the current study, this analysis was used to quantitatively demonstrate the shareability and uniqueness of microbial genera in each petroliferous region.

### Bivariate statistical analysis

The existence of possible statistically significant quantitative associations between pairs of microbial genera among petroliferous regions, for each sample type, was evaluated by Bivariate Statistics, using correlation analysis. Hellinger data transformation was performed (*Legendre & Gallagher, 2001*), and the non-parametric Spearman correlation test was used. Results were depicted as correlograms, for each sample type, where only significantly correlated microbial pairs were exhibited ($p \leq 0,05$). The Spearman correlation test and corresponding correlograms were performed using PAST v. 4.08 (*Hammer, Harper & Ryan, 2001*).

### Multivariate statistical analysis

A principal component analysis (PCA), which is a multivariate statistical analysis of ordination, was used to integratively explore all data. PCA summarizes the information of a very high number of original variables in a small set of statistical variables. In the current study, the interrelationships among sets of microorganisms were analyzed using the first two principal components (PC1 and PC2). After Hellinger transformation, data were analyzed by PCA, and visualized by two-dimensional scatter plots using ggplot2 v3.3.5 and factoextra v1.0.7 R packages (*Kassambara & Mundt, 2020*; *Wickham, 2011*).

### Functional prediction analysis

The functional prediction analyses were performed on all the available 16S rRNA amplicon metagenomes. Initially, SRA (Sequence Read Archive) or similar codes of all samples of all the selected documents were retrieved, considering at least one access code for each sample (or more than one in case of technical replicates). A total of 23 documents had to be discarded since the access codes were not provided. Therefore, all samples from a total of 35 documents were automatically extracted using the SRA-Toolkit (*Leinonen, Sugawara & Shumway, 2011*) for those with SRA codes, and manually for those with non-SRA codes.

The metagenomes were treated with VSEARCH v2.17.1 (*Rognes et al., 2016*), and pair-end metagenomes were joined (–fastq_mergepairs) in a same *FASTQ* file.

In case of having technical replicates in the samples, these were concatenated in the same *FASTQ* file to be treated as the same sample. For each unique *FASTQ* file, low-quality sequences were removed (–fastq_filter) and identical sequences were fused (–derep_fulllength), and all the resulting processed sequences were concatenated in a single *FASTQ* file. Using this single *FASTQ* file, singleton and chimeric sequences were removed, both without and with reference to the scripts –*uchime_denovo* and –*uchime_ref*, respectively, with the databases of SILVA alignments prepared with MOTHUR (https://mothur.org/wiki/silva_reference_files). Finally, operational taxonomic units or OTUs (–cluster_size) were generated in both *FASTA* and *BIOM* file formats.

Functional inference of metabolic pathways based on MetaCyc database (*Caspi et al., 2018*) was performed using the standard pipeline of the software PICRUSt2 (*Douglas et al., 2020*). The software STAMP v2.1.3 was used to infer significant statistical differences in the functional composition among all samples (*Parks et al., 2014*). The differences in functional composition were analyzed by pairs of sample types (IW-PW, IW-SD, IW-OIL, PW-SD, PW-OIL e SD-OIL), using Welch's t-test with the method of Benjamini–Hochberg FDR for the correction of multiple tests. A $p$-value > 0.05 was used as a threshold to discard statistically non-significant correlation between metabolic pathways.

## RESULTS

### Study selection

Among the 547 documents retrieved, 69 articles were selected because they could be directly used to respond to the main question of our study "What is the composition of bacterial and archaeal communities putatively associated with MIC found at distinct locations along the process of oil producing and transporting systems worldwide?" (Supplementary Material 5). Figure 1 depicts the PRISMA flowchart describing all stages of study selection, screening, and inclusion.

A total of 552 records from the Web of Science, Scopus, PubMed, and OnePetro databases were retrieved; and 99 documents without DOI from the Web of Science and Scopus databases were removed. The *format_input.py script* and the *remove_duplicates.py* removed duplicate records (59 articles), according to identical DOI or identical title, and six documents were not available and thus defined as *Not available*. The remaining 388 documents were then downloaded in *pdf* format. We found 309 original articles and 79 other types of documents (book chapters, books, reviews, and letters), which were excluded.

Initially, the remaining 309 documents were manually screened, based on their title, abstract, and entire documents. When necessary, the exclusion criteria described in Table 1 were applied: (1) oil refinery and petrochemical industry; (2) samples obtained from lab cultures, and (3) molecular analysis by Denaturing Gradient Gel Electrophoresis (DGGE) instead of metagenomics.

After this initial screening, 207 documents were excluded since they did not meet these eligibility criteria. Despite the application of the inclusion and exclusion criteria, a total of

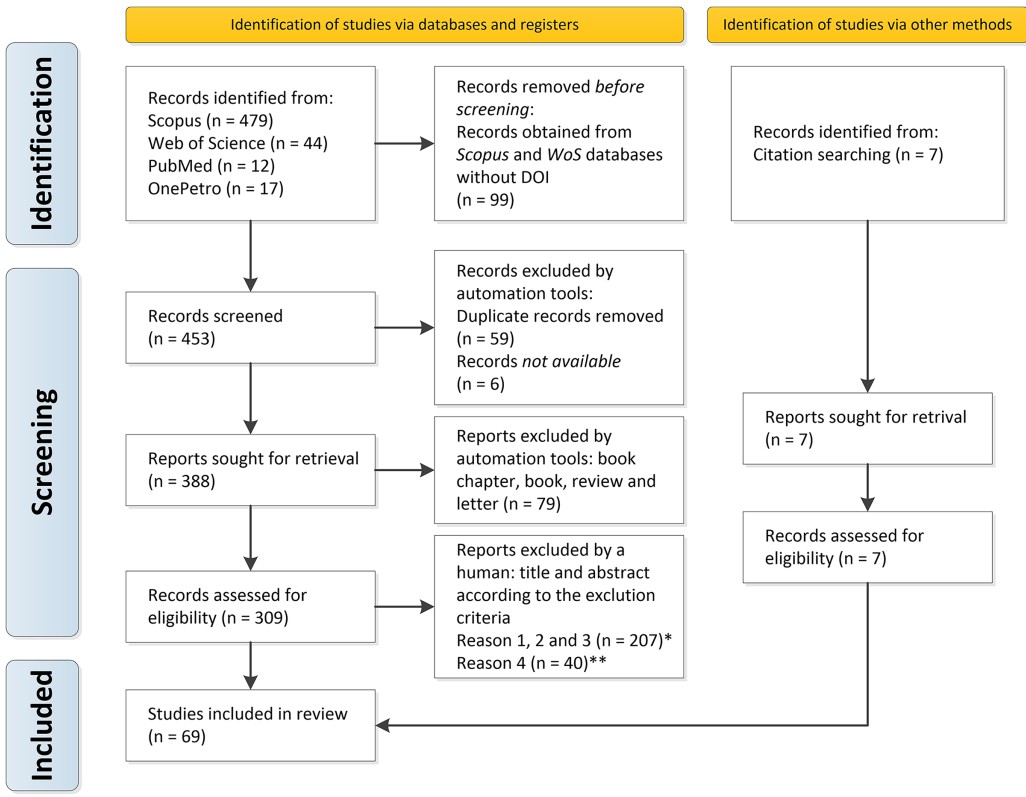

**Figure 1 PRISMA flowchart describing all the steps of inclusion of the documents in detail: selection, sorting, and inclusion.**

40 documents did not comply with these criteria in the data extraction stage. Thus, in the step of data extraction, they were excluded. A total of 7 articles considered important for the review, but which had not been found in the database search, were also included: these articles were identified from the bibliographic references of previously selected documents. Therefore, a total of 69 articles were included in our systematic review for data analysis and synthesis.

## Geographic location and sampling

Data was extracted and analyzed for each petroliferous region and each sample type. This method allowed the intrinsic analysis of the microbial profile of each variable (sample types and petroliferous regions). Figure 2 shows the distribution of petroliferous regions on the world map, and the number of samples for each region is depicted. A total of 368 samples were evaluated: 33 (8.97%) of IW, 25 (6.79%) of O, 36 (9.78%) of OW, 155 (42.12%) of PW, and 119 (32.34%) from SD (Supplementary Material 6). IW samples were analyzed from seven petroliferous regions, comprising a total of 18 articles, whereas PW was the sample type with the highest number of samples, described in 41 articles and distributed in seven petroliferous regions. O and OW samples were analyzed in eight and

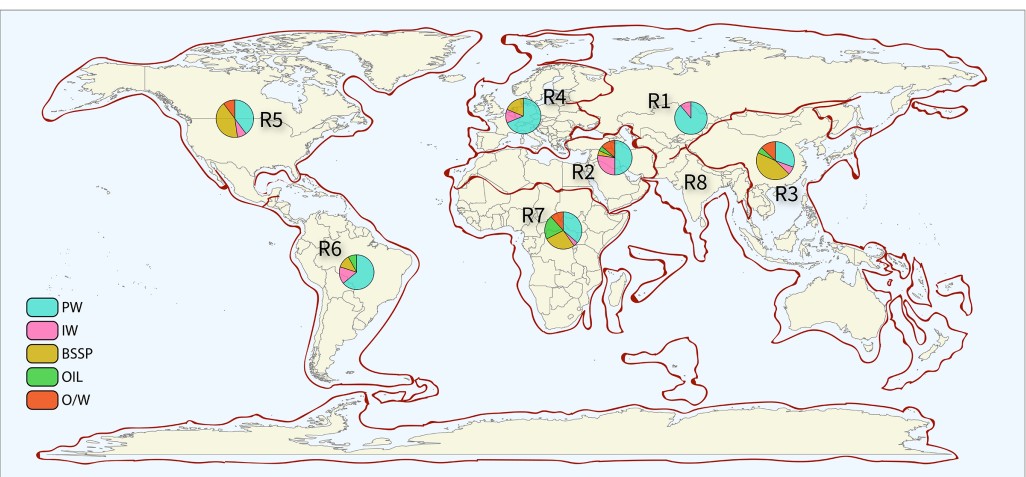

**Figure 2** **Sampling points by petroliferous region.** The pie charts indicate the type and number of samples. PW—produced water, IW—injection water, BSSP—biofilm/sediment/sludge/pig, Oil—Oil, and O/W—oil/water mixture. Petroliferous regions are indicated by: R1—Former Soviet Union, R2—Middle East and North Africa, R3—Asia-Pacific, R4—Europe, R5—North America, R6—Central and South America, R7—Sub-Saharan Africa and Antarctica, and R8—South Asia. All the sector (pizza-like) graphics and the legend were derived from authors' study, and the map of petroliferous regions (in the background) where the sector graphs and legend were plotted are completely redrawn from: USGS World Petroleum Assessment 2000, publicly available at https://pubs.usgs.gov/fs/fs-062-03/FS-062-03.pdf, using R and Illustrator by the authors.

nine articles, respectively. Finally, SD were characterized from 34 articles by both sample type and petroliferous regions.

The pie charts indicate the type and number of samples. PW—produced water, IW—injection water, BSSP—biofilm/sediment/sludge/pig, Oil—oil, and O/W—oil/water mixture. Petroliferous regions are indicated by: R1—Former Soviet Union, R2—Middle east and North Africa, R3—Asia-Pacific, R4—Europe, R5—North America, R6—Central and South America, R7—Sub-Saharan Africa and Antarctica, and R8—South Asia.

Table 2 shows the location, petroliferous regions, and countries of the oil fields reported in the selected articles. Samples from the 69 selected articles were distributed among seven petroliferous regions as follows: Region 1 (one article), Region 2 (six), Region 3 (18), Region 4 (eight), Region 5 (18), Region 6 (six), and Region 7 (12). No studies performed in the Region 8 were found. Moreover, each region displayed at least one country figuring in the ranking of the top 15 largest oil producers in the world (*IBP, 2020*).

## Description of molecular biology data and physicochemical features
### Molecular biology
To determine the composition of microbial communities and relative abundance of microbial lineages present in samples, the authors of selected documents extracted the community DNA and performed mainly amplicon (16S rRNA) metagenomics or, more rarely, shotgun metagenomics.

A total of five methods of metagenomic DNA extraction were identified: (1) commercial kits; (2) TRIzol-based; (3) phenol-chloroform-based; (4) QuickGene semi-automated

**Table 2 Number of articles selected by petroliferous region, including the country or region corresponding to the samplings.**

| Petroliferous region | Country or geographic region | Articles (*n*) |
|---|---|---|
| 1 | Kazakhstan—12th | 1 |
| 2 | Saudi Arabia—2nd | 4 |
| | Algeria | 1 |
| | Tunisia | 1 |
| 3 | Australia | 4 |
| | China—7th | 11 |
| | Japan | 1 |
| | New Zealand | 1 |
| | Papua New Guinea | 1 |
| 4 | Germany | 1 |
| | Denmark | 2 |
| | Norway—15th | 1 |
| | England | 1 |
| | North Sea | 3 |
| 5 | United States—1st | 12 |
| | Canada—4th | 4 |
| | Mexico—13th | 2 |
| 6 | Argentina | 1 |
| | Brazil—10th | 2 |
| | Colombia | 2 |
| | Gulf of Mexico | 1 |
| 7 | Nigeria—11th | 11 |
| | Guinea | 1 |
| Total | | **69** |

platform; and (5) Maxwell Automated Platform (Supplementary Material 7). Among the 69 articles, 57 used one of these methods, one (1) described its own methodology, six (6) referred to other authors, which are not included in the selected documents, and six (6) did not report the extraction method applied. The most cited methods in the selected articles were commercial kits (43) and the Maxwell automated platform (2). The articles mentioned commercial kits for DNA extraction from different sample types: soil (i); biofilm (ii); water (iii), and general (iv). In the present systematic review, the kits manufactured and indicated for DNA extraction from plants, animals, bacteria, algae and fungi were grouped and named "General". The biofilm kits were used for SD samples, and the water kits for IW and PW samples. The TRIzol reagent was referenced by an article for extracting DNA from PW samples.

Metagenomic DNA extracted from the samples was amplified using specific primers for the 16S rRNA gene from bacteria and archaea. In 14 studies, the universal pair of primers 926F (5′-AAA CTY AAA KGA ATT GAC GG-3′) and 1392R (5′-ACG GGC GGT GTG TRC-3′) were used to amplify the V6–V8 region of 16S rRNA genes from bacteria and

archaea. Considering sample types, these primers were used for amplification in 27 of them: IW (4), PW (8), SD (13), and O (2).

The articles reported Sanger (1st generation) and NGS (new generation) sequencing platforms. The Sanger method, carried out in the ABI PRISM 310, ABI PRISM 377, ABI PRISM 3730 (ThermoFisher, Waltham, MA, USA) sequencers, were described in six studies, while 454 (Roche, Basel, Switzerland) was reported in 23 studies, and MiSeq (Illumina, San Diego, CA, USA) in 35 studies. Article 212 (*Hernández-Torres et al., 2016*) cited the sequencing service provider Macrogen (Seoul, South Korea) (https://dna. macrogen.com/), and articles 245 (*Gittel et al., 2009*) and 249 (*Aktas et al., 2017*; *Duncan et al., 2009*) mentioned the Joint Genome Institute (JGI) of the United States Department of Energy (https://jgi.doe.gov/). Articles 42 (*Kamarisima et al., 2018*) 65 (*Sharma et al., 2018*) and 177 (*Lenhart et al., 2014*) did not describe the sequencing platforms that were used.

### Physicochemical features

In order to better understand the microbial communities, present in the studied samples, the physicochemical conditions of the sampled environments were analyzed, as well as the application (or not) of biocides and other possibly interfering chemical products. Microbial communities are dynamic, and the conditions prevailing in the systems will determine the community structure.

Therefore, in order to perform these analyses, data related to sampling point, temperature, pH value, salinity, sulfate and nitrate concentrations were collected, as well as data regarding the application of biocides and other chemical products (Supplementary Material 8). Among the included articles, 68.11% described the temperature, 63.77% the pH value, 56.52% the salinity, 68.11% the sulfate concentration, 28.98% the application of biocides and other chemical products, and 24.63% the nitrate concentration.

## Integrative analyses of microbiological data by sample type and petroliferous region

### Injection water

A total of 33 IW samples was characterized, Fig. 3. These analyses were carried out based on the relative abundances of microbial genera present in the IW samples.

The principal components (PC1 and PC2) of the ordination analysis (PCA) jointly explained approximately 40% of data variability (Fig. 3A). In addition, 17 genera that most influenced this variability were identified. Seven petroliferous regions were distributed in the four quadrants, as follows: group a (R3, R6 and R7), group b (R2), group c (R5) and group d (R1 and R4). Concerning the correlation analysis, only positive correlations were found (blue dots). Furthermore, groups of genera (G1 and G2) exhibiting strongly positive and statistically significant correlations were formed; however, with low relative abundance (Fig. 3B). There were no genera common to all regions but the R3 region had a higher number of unique microorganisms, followed by the R2 region (Fig. 3C).

Regions R1, R2, and R3 showed the highest number of genera, with high shareability and uniqueness, and relative abundances below 3% (0.34–2.94%)

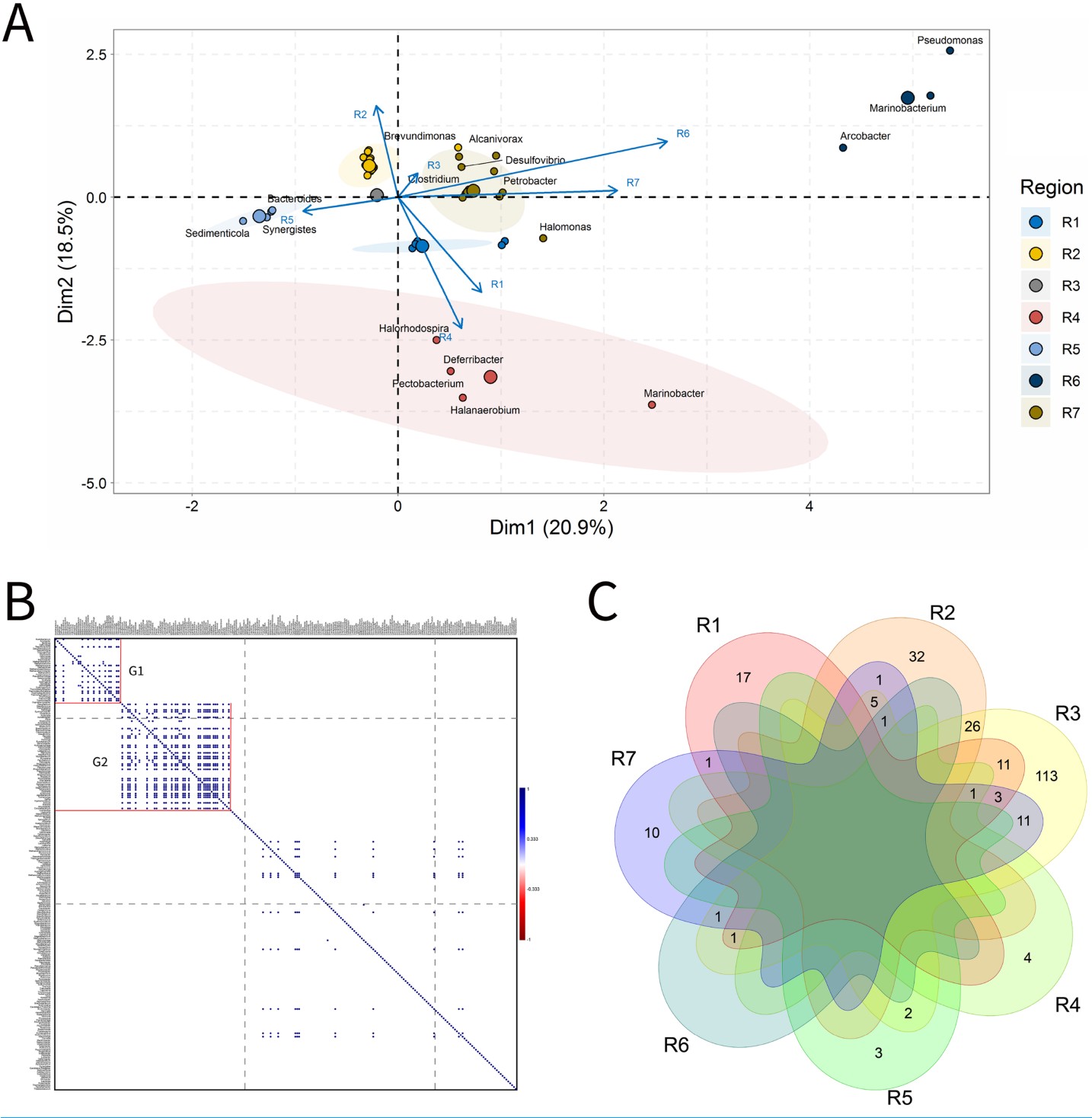

**Figure 3 Integrative analyses of microorganisms in Injection Water (IW).** (A) Principal components analysis; (B) Pairwise correlation between genera; (C) Shared and unique genera of the petroleum regions. This figure describes the integrative analyses of microorganisms in injection water (IW) (A) Principal components analysis; (B) Correlation of the microorganisms present in the injection water and (C) Venn diagram indicating the shared and unique bacterial and archaeal genera of the oil regions.

(Supplementary Material 9), Moreover, no statistically significant correlations were identified. The regions R4, R5 and R6 presented genera with high relative abundances (13.33–33.33%), and, among them, only the R5 region displayed statistically significant correlation.

Five genera were retrieved in the R4 region, and four of them were unique to this region; the genus *Halanaerobium* stood out for its/dominance, exhibiting the highest relative abundance (26.67%). In the R5 region, five genera were also detected, three unique and two shared with the R3 region. The dominant genera in this region were *Bacteroides*, *Synergistes*, and *Sedimenticola*, with high relative abundances (16.67–25%) (Supplementary Material 9); however, they are not known to participate directly in MIC processes. Moreover, the genera *Bacteroides* and *Methylophaga* were significantly positively correlated (Fig. 3B). The R5 region was considerably distinct from the other petroliferous regions (Fig. 3A). No significant negative correlations were detected.

The R6 region also displayed a very different community profile, with the largest number of genera that significantly influenced data variability. Three genera dominated the community: *Arcobacter*, *Pseudomonas*, and *Marinobacterium*. In the R7 region, the two most abundant genera were part of those also found in the R6 region: *Pseudomonas* and *Marinobacterium* (6.67%) (Supplementary Material 9), which justifies the proximity between these two regions (Fig. 3A and Supplementary Material 9). Nevertheless, the genera that most influenced data variability in this region were *Halomonas* (4.44%), *Petrobacter* (4.44%), *Alcanivorax* (4.44%), and *Desulfovibrio* (2.22%). There were statistically positive correlations between the nitrate-reducing bacteria (NRB) *Halomonas* and methanogens (MET) *Methanolobus* and *Methanosaeta* (Fig. 3B). Additionally, the genus *Halomonas* also positively correlated with the methanogens of the genera *Methanobacterium* and *Methanocalculus*.

### Produced water

PW samples are commonly evaluated to infer the microbial community present in subsurface environments (*Lipus et al., 2019*). For the analysis of PW, a total of 155 samples was characterized. In the ordination analysis, Fig. 4. PC1 and PC2 explained approximately 40% of the data variability with identification of 65 genera that most influenced this variability (Fig 4A). The seven petroliferous regions were distributed in two main groups: group b (R1, R2 and R5), and group c (R4 and R6) and those between these two groups (R3 and R7).

In the correlation analysis (Fig. 4B), both positive and negative statistically significant correlations between pairs of genera were retrieved. A large highly positively and significantly correlated group was formed, and this group comprised approximately 30 genera, some of which had considerably high relative abundances.

R3 region exhibited a higher number of exclusive genera (Fig. 4C). Additionally, two genera common to all regions were identified, which are representatives of the acid-producing (*Thermovirga*) and sulfate-reducing (*Desulfotomaculum*) functional groups.

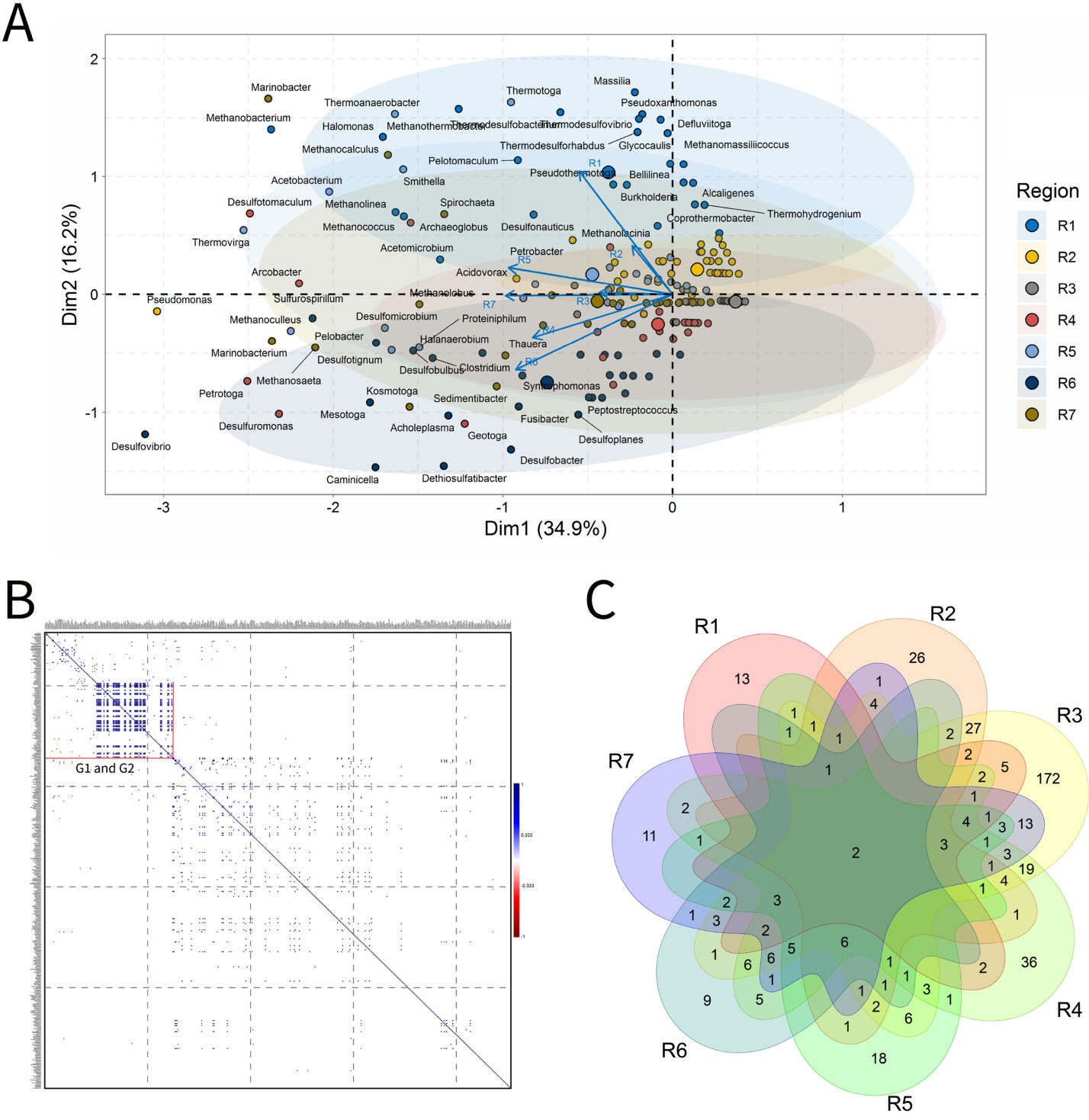

**Figure 4 Integrative analyses of microorganisms in Produced Water (PW).** (A) Principal components analysis; (B) Pairwise correlation between genera; (C) Shared and unique genera of the petroleum regions. This figure describes the integrative analyses of produced water oil regions (PW). (A) Principal components analysis; (B) Correlation of microorganisms present in produced water and (C) Venn diagram indicating the shared and unique bacterial and archaeal genera of petroleum regions.

The R1 region was composed of 45 genera, with 13 unique to it (Fig. 4C). All of these genera have a relative abundance between 0.58% and 4.07%. Statistically positive correlations were observed not only between genera within the same functional group, such as sulfate-reducing bacteria (SRB), MET and acid-producing bacteria (APB) but also between different functional groups (MET and SRB), (MET and iron-reducing bacteria—IRB) and (sulfur-reducing bacteria—$S^0$RB and APB) (Fig.4B). Among the statistically negative correlations, there were also correlations within the same functional group (MET) and between different functional groups (aerobic bacteria of the class Betaproteobacteria, NRB, SRB, and $S^0$RB), (MET and APB) and genera of the G1 group (*Brevundimonas*, *Sphingomonas*, and *Tistrella*) from the R2 region, with an abundance of less than 3.0%, aerobes of the class Alphaproteobacteria (Fig. 4B and Supplementary Material 9) (Supplementary Material 9). Those genera with the highest relative abundance comprised *Defluviitoga* (4.07), *Thermodesulfovibrio* (4.07%), *Pelotomaculum* (4.07%), *Desulfotomaculum* (4.07%) (Supplementary Material 9).

The R2 region display important genera of the phylum Proteobacteria that correlated significantly and positively, such as *Petrobacter* and *Pseudomonas*, which are NRB. Negative correlations were also retrieved between NRB and APB of the genera *Pseudomonas* and *Acetomicrobium* (2.33%), respectively, from the R1 region.

The R3 region was composed of 322 genera, of which 172 were unique to it (Fig. 4C), and presented only genera with relative abundances below 2.0%. This region exhibited fourteen genera with an abundance above 1.0%, and those with the highest abundance values were *Pseudomonas* (a hydrocarbon-degrader NRB) and *Methanosaeta* (MET) with 1.61%, (Supplementary Material 9). Among the diverse genera that were identified, a total of 18.75% were constituted by genera of the class Deltaproteobacteria, which contains many SRB, and 18.18% of the phylum Euryarchaeota, in which all representatives are MET.

The R4 region was composed of 123 genera, with 36 unique to it (Fig. 4C). The genera that effectively influenced data variability were: sulfidogenic archaea, *Archeoglobus* (5.00%); NRB, *Arcobacter* (2.73%); SRB, *Desulfotomaculum* (4.09%) and *Desulfuromonas* (3.64%); $S^0$RB, *Petrotoga* (3.18%) and *Geotoga* (2.27%), (Fig. 4A and Supplementary Material 9). In addition, there were significant positive and negative correlations between some genera (Fig. 4B). Two sulfidogenic genera *Desulfuromonas* and *Geotoga* were positively correlated. *Desulfuromonas* was also positively correlated with another sulfidogenic genus abundant in the R6 region, *Desulfovibrio* (SRB). These sulfidogenic genera (*Desulfuromonas* and *Geotoga*) were correlated negatively with the thermophilic and hydrogenotrophic methanogen *Methanothermobacter*, which was the one that most contributed to explain data variability in the R1 region. In turn, the genus *Geotoga* was also correlated with other genera, highlighting negative correlations with the thermophilic *Thermoanaerobacter*, an iron-reducing genera (*Slobodkin et al., 1999*), and $S^0$RB *Thermotoga* from the R5 region (*Davey et al., 1993*). A negative correlation was also found between *Petrotoga* and the thermophilic fermentative and aerobic NRB *Halomonas* from the R1 region.

The R5 region was composed of 72 genera and, among them, 18 unique to that region (Fig. 4C). The genera that significantly influenced the ordering of the R5 region were: *Methanoculleus* (5.21%), *Acetobacterium* (4.86%), *Desulfomicrobium* (4.17%), *Smithella* (3.12%), *Thermotoga* (2.78%), *Thermoanaerobacter* (2.43%), and *Proteiniphilum* (2.08%) (Fig. 4A and Supplementary Material 9). The genera in this region were positively correlated (Fig. 4B). The APB *Acetobacterium* and *Smitella* were positively correlated between them and also with other genera from different petroliferous regions. *Acetobacterium* positively correlated with *Methanocalculus*, a hydrogenotrophic methanogen abundant in the R7 region. Additionally, *Smitella* was positively correlated with *Desulfonuticus*, an elemental sulfur or thiosulfate reducer with high abundance in the R1 region. There were also correlations between thermophilic anaerobic genera with less than 3.0% abundance, such as *Thermoanaerobacter*, with iron reducers, and *Thermotoga*, an S⁰RB and APB. These aforementioned genera were also correlated with MET, S⁰RB, and APB.

The R6 region presented 59 genera, with nine unique to this region (Fig. 4A). Several genera that significantly influenced the ordering of this region were represented; however, those with the highest relative abundance were: *Desulfovibrio* (5.45%), *Desulfobacter*, *Caminicella*, *Mesotoga*, and *Dethiosulfatibacter* (all these genera with 4.55%). This region has statistically significant positive and negative correlations (Fig. 4B). The genera SRB *Dethiosulfatibacter* and *Desulfobacter* (Lien & Beeder, 1997; Takii et al., 2007) were positively correlated with each other and with the genus *Geotoga* (S⁰RB) in the R4 region. Additionally, *Dethiosulfatibacter* and *Desulfobacter* were also negatively correlated with *Marinobacter*, a NRB that was dominant in the R7 region. The S⁰RB *Caminicella* was positively correlated with other S⁰RB genera in the R4 and R7 regions; and negatively with the thermophilic S⁰RB genera, the most relevant in the R5 region, as well as with thermophilic MET *Methanothermobacter* in the R1 region. The SRB *Desulfovibrio*, displayed significant positive correlations with NRB and other SRB genera that are resistant to pollutants, such as *Sulfurospirillum* from the R7 region (Goris & Diekert, 2016) and S⁰RB genera from the R4 region. Furthermore, it also negatively correlated with the aerobic genus *Massilia*, which is influential in the R1 region.

The R7 region was composed of 84 genera, 11 of which were unique to it (Fig. 4A). The most important genera that were central in the ordination analysis of this region were: *Thauera* (3.05%), *Petrobacter* (3.06%), *Methanobacterium* (3.07%), *Methanocalculus* (3.05%) (Fig. 4) and *Methanosaeta* (3.32%), *Spirochaeta* (3.60%), *Thermanaerovibrio* (3.88%), *Marinobacter* (4.16%), *Pseudomonas* (4.16%), *Methanolobus* (4.43%), and *Marinobacterium* (4.71%) (Fig. 4A and Supplementary Material 9). Four groups of positively correlated genera were observed (Figs. 4B and 4C). The first one included the APB *Spirochaeta*, and the MET *Methanococcus*, (with higher abundance in the R1 region) and the SRB *Desulfomicrobium*, which was the most relevant in the R5 region. The second one comprised the MET *Methanocalculus*, and the hydrocarbon-degrader NRB *Halomonas* (R1) and *Marinobacter* (R7). The third one was composed of the MET *Methanobacterium* and the APB *Acetobacterium* (R1). Finally, the fourth one, which displayed correlation between the NRB *Marinobacterium* and the MET *Methanoculleus*,

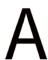 PeerJ

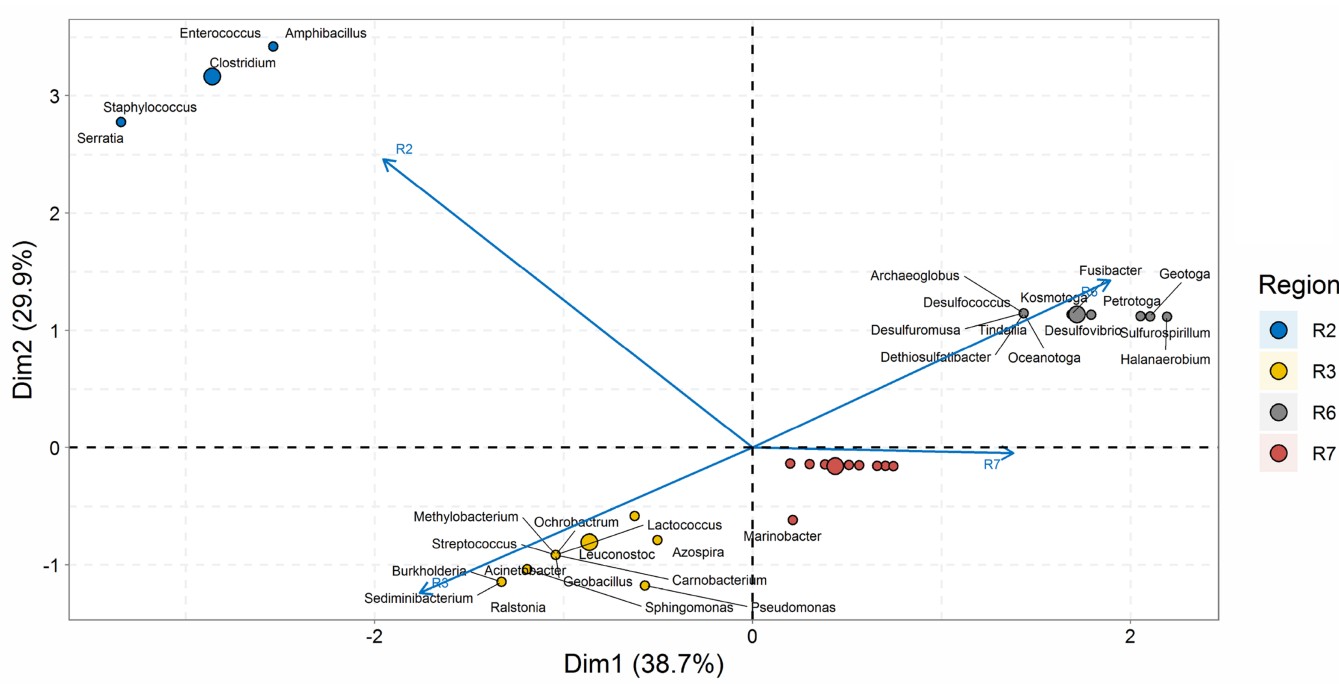

A

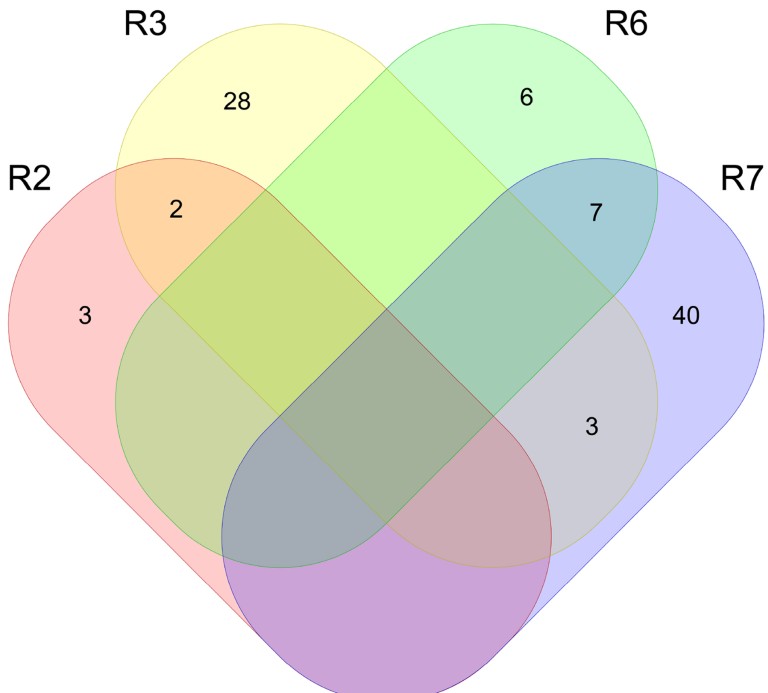

B

**Figure 5 Integrative analyzes of oil samples in oil fields.** (A) Principal component analysis and (B) Venn diagram indicating the shared and unique bacterial and archaeal genera of petroleum regions. Full-size 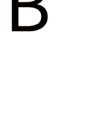 DOI: 10.7717/peerj.14642/fig-5

the most relevant in the R5 region, *Methanosaeta* and *Methanolobus*, the most relevant in this R7 region. Regarding the negative correlations, the genus *Marinobacterium* was correlated with the aerobic genus *Massilia*, which was the most relevant in the R1 region.

### Oil (O) and Oil/Water

In the present study, 25 O samples and 36 OW samples were analyzed; and both did not exhibit genera that were statistically significantly correlated (Fig. 5). In the ordination analysis, PC1 and PC2 explained almost 69% of the data variability (Fig. 5A). This type of sample was represented by 33 most relevant genera, grouped in four distinct groups: group a (R6), group b (R2), group c (R3), and group d (R7). It is observed that the vectors of regions R3 and R6 were in opposite directions, which indicates possible negative correlations between the genera of these regions, but these correlations were not significant. Genera shared by all the petroliferous regions were not identified, and the R3 region had a high number of unique genera when compared to the other ones (Fig. 5B).

The R2 region (Fig. 5A), had a greater distance in relation to the other regions. A total of 90% of the genera was from Bacilli and Clostridia classes of the Firmicutes phylum, such as *Amphibacillus*, *Clostridium*, *Enterococcos*, *Serratia*, and *Staphylococcus* (Fig. 4C). These classes contain APB, such as *Clostridium*, that accounted for a relative abundance of 20% (Supplementary Material 9). In the R3 region, 14 genera were considered those that most contributed to the ordination of this region; however, the genera with the highest relative abundance were *Ralstonia*, *Burkholderia*, *Sediminibacterium*, and *Pseudomonas* (all of them with 6.49%) (Fig. 4A and Supplementary Material 9).

The R6 region had 13 genera, of which 6 are unique (Fig. 5B), and all genera of this region were represented in the ordination analysis (Fig. 5B) displayed a relative abundance of 7.69%. In this region, a clear dominance of SRB and high abundance of APB and $S^0RB$ was observed. On the other hand, in the R7 region, only one genus that influenced the data variability was highlighted, *Marinobacter* (NRB), with a relative abundance of 4.58%. Regarding the analysis of shareability and uniqueness, this region exhibited 50 genera with 40 unique ones (Fig. 5B).

The results of the integrative analyses related to the OW samples are depicted in Fig. 6. In ordination analysis, PC1 and PC2 explained approximately 69% of the data variability (Fig. 6A), and a total of 47 genera were the most relevant for the ordination in three groups: group a (R3), group b (R2) and group c (R5 and R7). Unlike the other samples types (IW, PW), there were OW data only for four petroliferous regions (R2, R5, R3 and R7) (Fig. 6B). Moreover, genera shared with all regions were not identified, and the R3 region displayed a high number of unique genera when compared to the other regions.

The R2 and R3 regions were found to be distant from each other and from the R5 and R7 regions. In the R2 region, 16 genera were described, and 6 of them were unique to this region (Fig. 6B). The genera with the highest relative abundances were the sulfidogens *Pelobacter*, *Petrotoga*, *Acetomicrobium*, *Clostridium*, *Flexistipes*, *Thermoanaerobacterium*, and *Thermovirga*.

The regions R2 and R3 (Fig. 6B) were far from each other and from the other regions, which may indicate that they have unique features. In the R2 region, 16 genera were

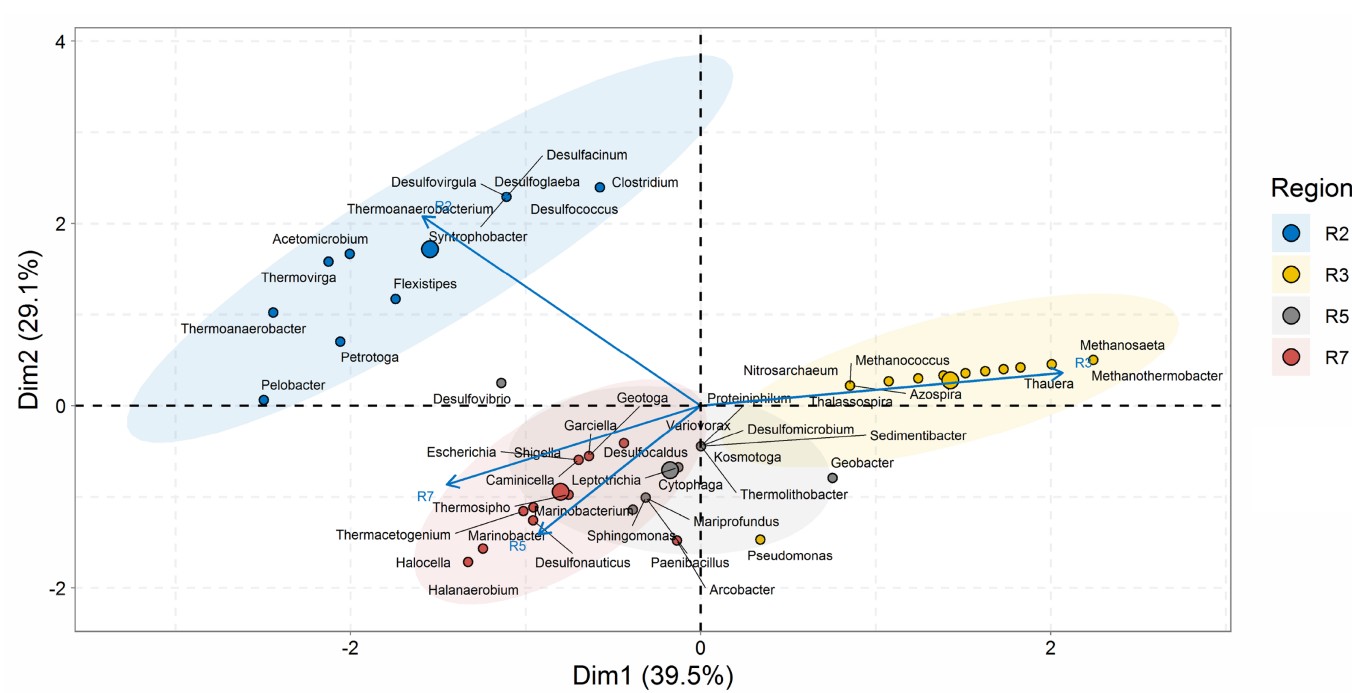

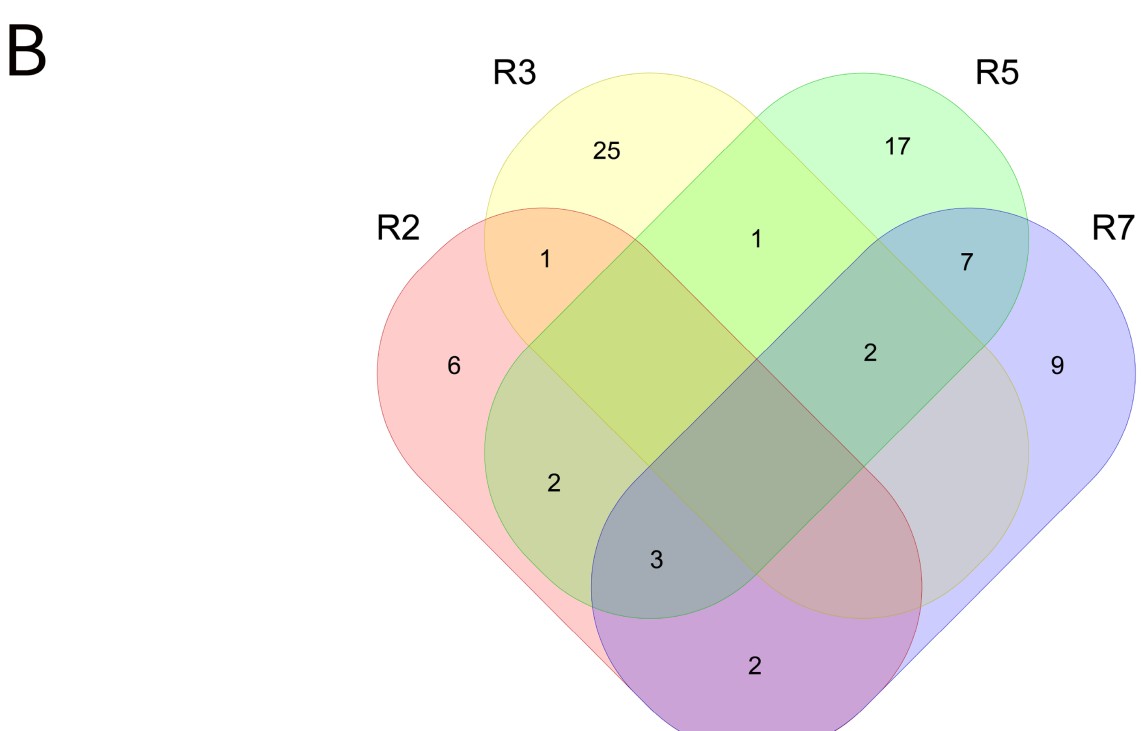

**Figure 6 Integrative analysis of oil and water mixture samples (OA).** (A) Principal component analysis and (B) Venn diagram indicating the shared and unique bacterial and archaeal genera of petroleum regions.

retrieved, and 6 of them were unique (Fig. 6A). Relative abundances varied between 2.63% and 7.89%, and only *Desulfovibrio* (2.63%) exhibited an abundance below 3.0% (Supplementary Material 9). In the R3 region, 29 genera were described, and 25 of them were unique (Fig. 6B). Nine genera were the most relevant to data variability (Fig. 6A), including representatives of the functional groups NRB, $S^0$RB, and MET, with *Methanosaeta* and *Methanothermobacter* (8.33%) as the most abundant genera. The R5 and R7 regions were relatively close to each order in the ordination analysis when compared to the other regions (Fig. 6A). These regions displayed genera with relative abundances between 1.14–5.68% (Supplementary Material 9).

The R5 region presented 32 genera, with 17 of them unique to this region (Fig. 6B). The most important genera for the ordination of the R5 region werethr IRB *Geobacter* and the SRB *Desulfovibrio* (Fig. 6A), besides 17 genera with an abundance greater than 3.0%, including those from NRB (Supplementary Material 9). In the R7 region, 23 genera (of which six unique ones) were described (Fig. 6B): there was dominance of APB (*Pelobacter* e *Acetomicrobium*) NRB (*Marinobacter* e *Marinobacterium*), thiosulfate-reducing bacteria —TRB, and $S^0$RB (*Thermacetogenium*, *Thermosipho*, *Geotoga*, *Desulfonauticus*), as well as a smaller number of SRB were also detected, as well as in the R5 region (Fig. 4A).

## Solid deposits

For BSLP, a total of 155 samples was retrieved and analyzed, Fig. 7. In the ordination analysis, PC1 and PC2 explained 52.5% of data variability, and the petroliferous regions were distributed in two groups: group b (R2 and R5) and group c (R3, R4, R6 and R7) (Fig. 7A). In the correlation analysis, there were significant positive correlations and a small number of significant negative correlations (Fig. 7B). Nonetheless, extensive groups of correlated genera were not formed, as happened in IW and PW samples. There were representatives in six petroliferous regions: R2, R3, R4, R5, R6 and R7 (Fig. 7C). Additionally, only one genus was shared among all regions: the NRB *Marinobacter*. As occurred for IW, PW and OW samples, the R3 region of SD exhibited the highest number of unique microorganisms.

The R2 region (group b) was segregated from all the other regions in the ordination analysis (Fig. 7A), and showed only 10 genera, with four of them unique to this region (Fig. 7B). The NRB *Chromohalobacter*, (Aguilera et al., 2007) and the TRB *Halanaerobium* were significantly positively correlated (Fig. 7C). On the other hand, the NRB *Halomonas* was negatively correlated with the MET *Methanosarcina*, which was more abundant in the R6 region: these genera showed a relative abundance equal to 10% (Supplementary Material 9).

In the ordination analysis, the R3, R4, R5, R6 and R7 regions were relatively closer to each other than with R2 region (Fig. 7A). The R3 region had 173 genera, of which 117 unique to it (Fig. 7B), with an abundance of less than 3.0%. This region showed genera from the functional groups: hydrocarbon-degraders, APB and IRB, and the most abundant was *Pseudomonas* (2.78%), followed by the IRB *Shewanella* (2.20%), and APB, such as *Proteiniphilum* (2.22%) (Supplementary Material 9).

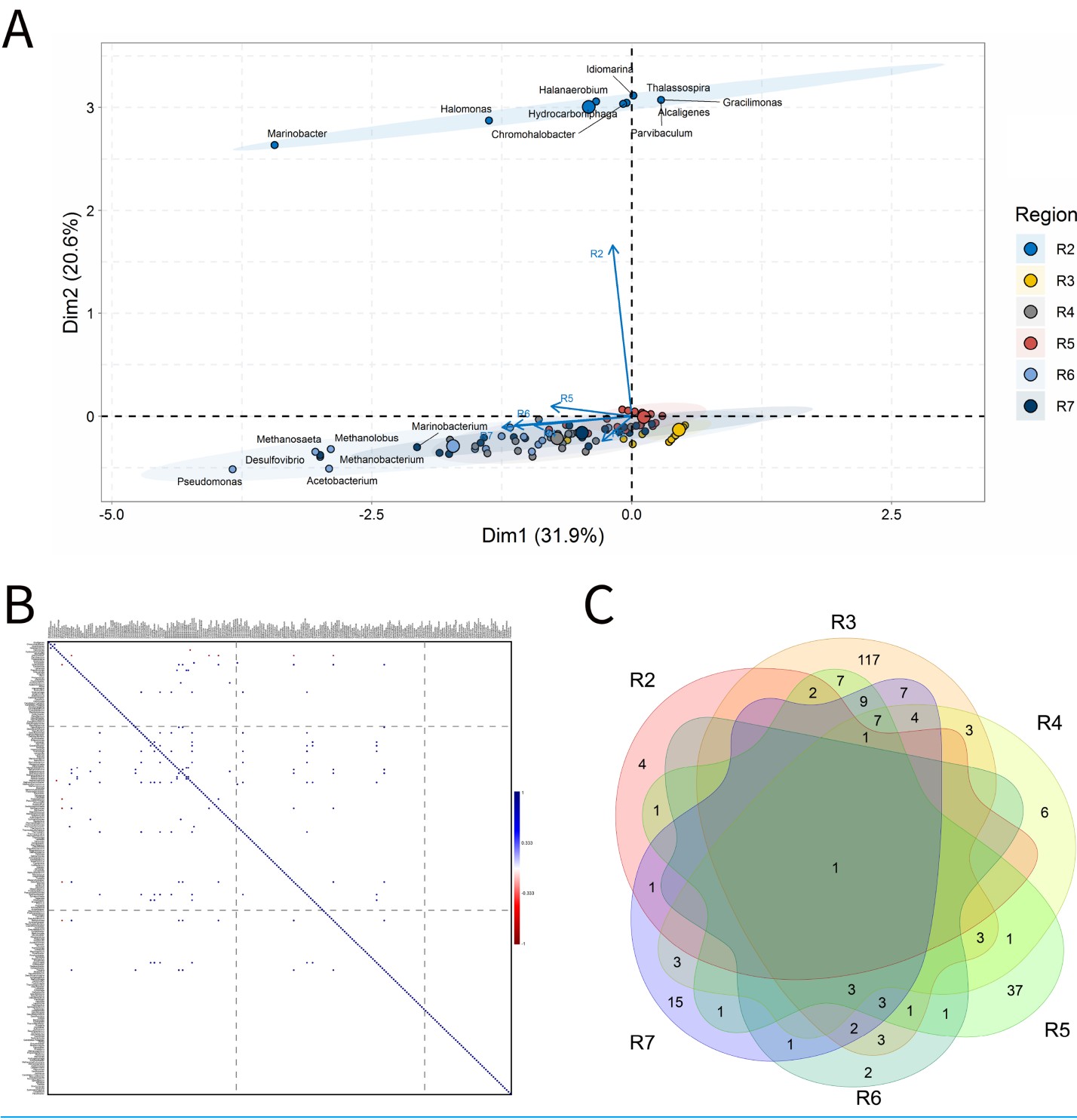

**Figure 7 Integrative analyses of microorganisms in solid deposit samples (BSLP).** (A) Principal components analysis; (B) pairwise correlation between genera; (C) shared and unique genera of the petroleum regions. This figure describes the integrative analyses of solid deposit samples (BSLP) in oil fields. (A) Principal components analysis; (B) correlation of microorganisms present in produced water and (C) Venn diagram indicating the shared and unique bacterial and archaeal genera of petroleum regions.

The R4, R5, R6, and R7 regions displayed similar genera and shared many functional groups. The R4 region had 32 genera, of which six unique to it (Fig. 7B), with relative abundances between 1.61% and 6.45%. This region was dominated by sulfidogenic thermophilic genera (S$^0$RB and SRB), NRB, and MET, with greater abundance of *Thermococcus* and *Desulfonatrospira* (6.45%), followed by the *Methanothermococcus* (4.84%).

There were statistically significant positive correlations between the MET *Methanothermobacter* and genera with an abundance above 3% in the NRB, APB, S$^0$RB and IRB functional groups. In turn, the R5 region had 78 genera, 37 of which are unique (Fig. 7B). The relative abundance varied between 0.52% and 4.17%, with *Pseudomonas* (4.17%) as the most abundant, followed by *Desulfomicrobium*, *Desulfovibrio*, and *Methanosaeta* (3.65% each).

The R6 region exhibited 17 genera, with two unique to it (Fig. 7C). This region displayed genera associated with NRB, SRB, APB, and MET functional groups (Figs. 7B and 7C). Two groups were significantly positively correlated: the first one between MET *Methanococcus* and the APB *Acetobacterium*; and the second one between MET *Methanolobus* and *Methanosaeta* and the hydrocarbon-degrader NRB *Pseudomonas*. Furthermore, the later correlated positively with the most abundant genera in the R7 region: the MET *Methanobacterium*.

The R7 region had 58 genera, of which 15 unique to it (Fig. 7B), with relative abundances between 0.38% and 6.82%. Six main genera were detected, and only the S$^0$RB *Thermanaerovibrio* did not significantly correlate with the other genera. In the R7 region, there were positive correlations between genera within the region and with genera of the R4 region, which had an abundance greater than 3.0%. The MET *Methanoculleus* and *Methanocalcullus* correlated positively each other and also with *Thauera* (3.41%), anaerobic hydrocarbon degrader. The NRB *Marinobacterium* and the S$^0$RB *Kosmotoga* were positively correlated with each other and also with APB *Sedimentibacter*. Moreover, *Marinobacterium* and *Kosmotoga* (3.23%) also correlated with genera of S$^0$RB, APB, and MET groups in the R4 region (*Desulfuromonas*, *Anaerophaga Methanothermobacter* and *Thermosipho*). The S$^0$RB *Kosmotoga* correlated with the APB *Spirochaeta* and *Geoalkalibacter* (3.23%) from the R4 region. *Geoalkalibacter* is strictly anaerobic, and there are species that can utilize Fe(III), Mn(IV), nitrate, elemental sulfur and trimethylamine *N*-oxide as terminal electron acceptors, and a wide variety of organic acids as electron donors. The most abundant genus is *Methanobacterium* (6.82%), followed by *Methanolobus* and *Methanosaeta* (5.68%). This region was dominated by the hydrogenotrophic methanogens *Methanobacterium*, *Methanocalculus*, *Methanoculleus*, and *Methanosaeta*.

## Functional prediction

Among the 16S rRNA metagenomes recovered from different sample types, a total of 440 metabolic pathways was identified in Metacyc at the highest classification level using PICRUSt2. Removal of statistically non-significant metabolic pathways ($p \geq 0.05$) resulted in a total of 167 pathways, Fig. 8. Subsequently, these pathways were manually classified according to Metacyc level 2 in order to facilitate understanding, totaling 26 significant

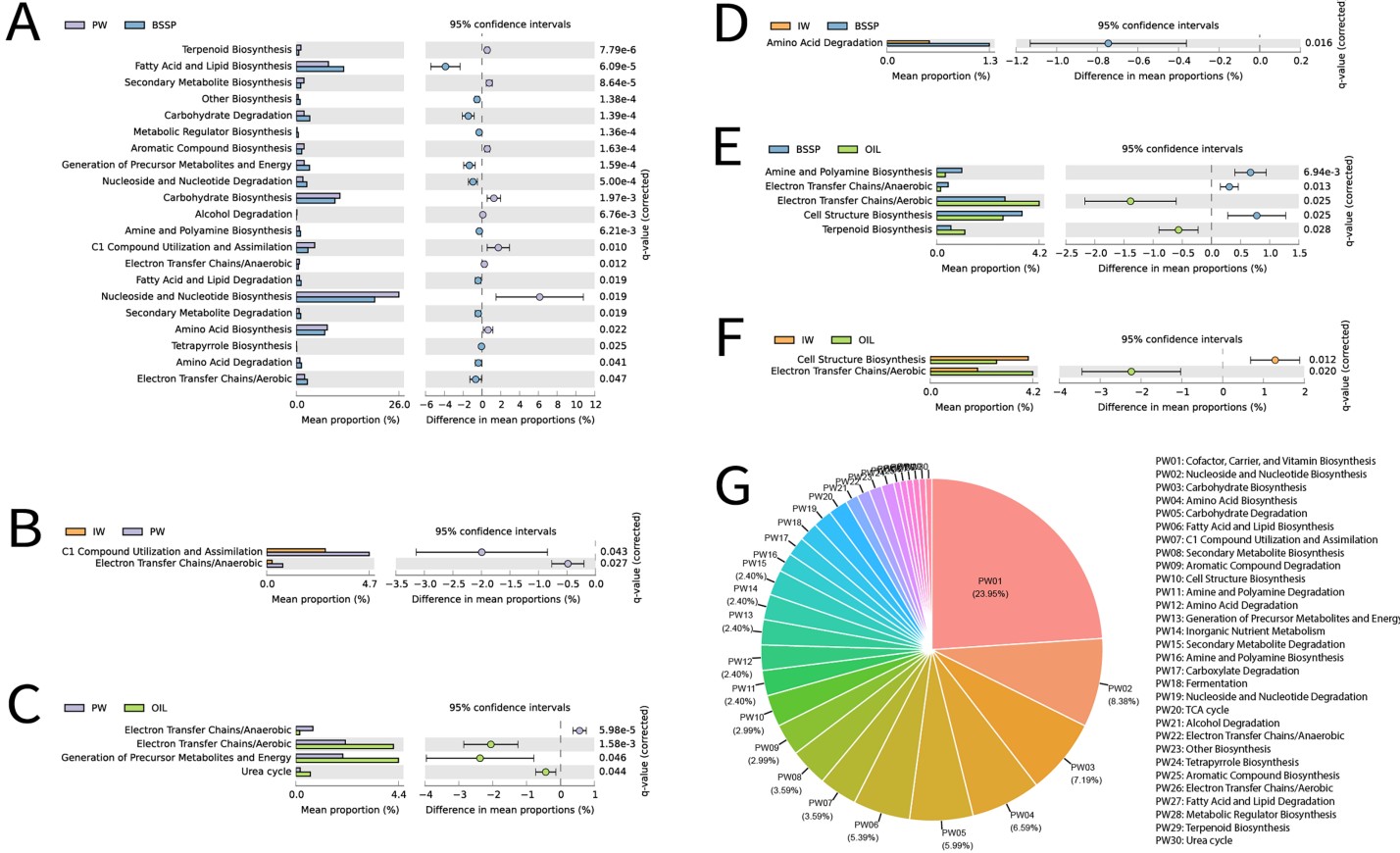

**Figure 8** (A–G) Extended error bar graph for comparing analysis module of two groups of MetaCyc functional data predicted by PICRUSt2, and the pie graph illustrating the predicted functions in the different sample types.

metabolic pathways. The most frequent functions in these samples were (in this order): (1) Cofactor, Carrier, and Vitamin Biosynthesis, (2) Nucleoside and Nucleotide Biosynthesis, (3) Carbohydrate Biosynthesis, (4) Amino Acid Biosynthesis, (5) Carbohydrate Degradation, and (6) Fatty Acid and Lipid Biosynthesis (Fig. 8G).

Extended error bar graph was used for the comparison between two samples. The bar graphs on the left display the average proportion of each MetaCyc pathway, while the dot plots on the right show the differences in the average proportions between two samples: A —produced water (PW) and solid deposits (BSSP); B—injection water (IW) and produced water (PW); C—produced water (PW) and oil; D—injection water (IW) and solid deposits (BSSP); E—solid deposits (BSSP) and oil; F-injection water (IW) and oil. Only predicted functions with $p < 0.05$ are shown; G—pie chart of predicted functions and substrate types.

A pairwise comparative analysis of the metabolic prediction of the different sample types were carried out to identify the greater or lesser similarity between sample types and functional categories from the different petroliferous regions (Fig. 8). Since PW and SD retrieved a higher number of 16S rRNA metagenomes when compared to the other sample types, the combination of these sample types (PW and SD) also exhibited higher amounts

of common functions (21), followed by the combination of SD and O (five), PW and O (four), IW and PW (two), IW and O (two) and IW and SD (one) (Figs. 8A–8F).

For PW and SD, Nucleoside and Nucleotide Biosynthesis was the most expressive functional category, followed by Fatty Acid and Lipid Biosynthesis, Carbohydrate Biosynthesis, and Amino Acid Biosynthesis (Fig. 8A). The most relevant difference between these samples was the higher abundance of Nucleoside and Nucleotide Biosynthesis in PW and Fatty Acid and Lipid Biosynthesis in SD. In contrast, in the combination of PW, IW and O, the metabolic pathways were quite distinct. Between PW and IW, C1 Compound Pathway Utilization and Assimilation stood out, with higher abundance in PW (Fig. 8B). In relation to PW and O, the most expressive pathways were Electron Transfer Chains/Aerobic, and Generation of Precursor Metabolites and Energy, which were more abundant in O samples (Fig. 8C). Regarding the combinations of SD with IW, there was greater prominence of the Amino Acid Function Degradation (Fig. 8D). On the other hand, in the combination between SD and O, the Electron Transfer Chains/Aerobic were observed with higher abundance in O samples, and Cell Structure Biosynthesis was more abundant in SD (Fig. 8E). These functional categories also stood out in the combination of IW and O: Cell Structure Biosynthesis was more abundant in IW whereas Electron Transfer Chains/Aerobic was more abundant in O (Fig. 8F).

## DISCUSSION

This study evaluated the microbial profile of samples associated with the oil industry (PW, IW, OW, OIL, and SD) in seven petroliferous regions, comprehending several countries all over the world. Many of these countries figure in the ranking of the top 15 oil producers in the world (IBP, 2020) and each petroliferous region has at least one country classified in this ranking (Table 2). This demonstrates, in general, that the current systematic review constitutes a representative sample of the world scenario, referring to the composition and abundance of microorganisms present in petroliferous regions. The countries with the highest number of articles were the USA, China, and Nigeria. USA is the largest oil producer in the world, China occupies the seventh place, and Nigeria, the eleventh place. Thus, this scenario can corroborate the growing interest of these countries in research related to the oil sector.

The PW and SD sample types were the most numerous, with 155 (42.12%) and 119 (32.34%) samples, respectively, out of a total of 368. PW is one of the most common sources of microbiological samples (Rachel & Gieg, 2020), mainly because it is considered a representative sample of the oil field, for having had direct contact with the oil reservoir and injection water, making it possible to predict circulating (planktonic) microorganisms in the water column, besides the logistic facilitation if compared to collections carried out directly in subsurface. On the other hand, SD allows the analysis of sessile microorganisms, which form biofilms, responsible for localized corrosion (Carrascosa et al., 2021). These samples are obtained through pig residues and/or swabs of metallic coupons, commonly used in oil industries to monitor corrosion rates (Sliem et al., 2021). Furthermore, pigs are the only equipment that can access the entire length of a pipeline, for cleaning and inspection purposes, and can be complemented by metallic coupons, which are fixed at

points (located at the beginning and end of pipelines due to logistic facilitation) on the walls of the ducts to be exposed to internal conditions. Nevertheless, as many ducts are not possible to be inspected by pigs, it is not always possible to obtain this type of sample in oil industries.

Regarding molecular biology methods, we detected that 62.3% of the documents performed DNA extraction using commercial kits specific for each sample type. This is justified by the complexity of the material, as they may have elements that influence cell lysis, isolation of nucleic acids or polymerase activity (*Oldham et al., 2012*; *Wilson, 1997*). For DNA sequencing, 84% of documents used NGS platforms. Regarding the data obtained from physicochemical characteristics and application of chemical products, we realized that, although usually most of the articles offered information on at least one parameter, there was no priority in reporting this information, or there was even a lack of such information. This scarcity of such kinds of data may have occurred due to the difficulty to obtaining them.

Concerning the IW, PW and SD samples, the region R3 presented the smallest number of genera able to explain data variability (Figs. 3C, 4C, 5B, 6B and 7C), with relative abundances inferior to 3.0%. Additionally, in all sample types, except for O, there was a higher number of unique genera (Figs. 3C, 4C, 5B, 6B and 7C). One of the probable explanations for this pattern would be the high number of studies (18) distributed just in five countries with a greater prominence in China, with 11 articles (Table 2). This likely favored data heterogeneity, and thus a high variability of the physicochemical characteristics and application of chemical products. Therefore, this scenario resulted in an increase in the diversity of organisms with different metabolisms (*Liu et al., 2019*) evaluated the changes in microbial diversity in petroleum exploration, and concluded that environmental variation (temperature, $O_2$ concentration, pH value, and salinity) (Supplementary Material 8) alters the diversity of microbial communities during oil production.

In IW samples, the R6 region displayed the largest vector, indicating greater importance in explaining data variability, highlighting the genera *Arcobacter*, *Pseudomonas*, and *Marinobacterium*. The regions R6 and R7 were the closest ones, which suggests a higher similarity between the variables, sharing the genera *Pseudomonas* and *Marinobacterium*. The R4 also highlighted, exhibiting the *Marinobacter* genus, which is present in relative abundances above 2% in the R1 and R7 regions.

The genus *Arcobacter* participates in elemental sulfur oxidation, nitrogen metabolism, and iron and manganese reduction. Moreover, many studies suggest that these bacteria may not be native of oil reservoirs, but artificially introduced (*Bedoya et al., 2021*). Bacteria of the genus *Pseudomonas* are found in diverse terrestrial and aquatic environments, including oil reservoirs (*Dellagnezze et al., 2016*). *Pseudomonas* is a versatile, heterotrophic, facultative aerobic genera with evidence of fermentation (*Jeon, Yi & Park, 2014*; *Ramos, 2018*) and capable of degrading high fractions of crude oil constituents in the presence of $O_2$ but it can also prevail under anaerobic facultative conditions in oil/water mixtures (*Bedoya et al., 2021*). On the other hand, the genus *Marinobacterium* is facultative anaerobic (*Conlette & Nnameka, 2018*) and consists of nitrate-reducing species. In turn,

*Marinobacter* species, besides being nitrate reducers, are halophilic, mesophilic, and tolerant to pH variations from 6.0 to 9.5; they are aerobic with non-fermentative metabolism and can grow anaerobically in the presence of nitrate and acetate, but do not grow in the presence of glucose. They use nitrate as terminal electron acceptor, generating $N_2$, and can utilize acetate and other organic acids as the sole carbon source. Furthermore, they are capable of degrading a wide range of hydrocarbons (aliphatic and aromatic) in anaerobic conditions (*Duran, 2010*).

The dominance of the NRB group may have been favored by the application of biocides. *Bedoya et al. (2021)* evaluated the microbial community and biocide resistance profile in PW and IW of an oil reservoir in Colombia: the injected water was treated with biocides glutaraldehyde and tetrakis (hydroxymethyl) phosphonium sulfate (THPS). The authors reported a dominance of the genera *Arcobacter* and *Pseudomonas* found more than 20 genes associated with antibiotic and biocide resistance. One of these genes was *opr*N (outer membrane efflux protein), which encodes a multidrug efflux pump related to glutaraldehyde resistance (*Vikram, Bomberger & Bibby, 2015*). They concluded that the biocides applied in the IW probably had induced significant changes in the community structure, since long-term treatment with biocides can induce a selection process, which reduced diversity and strongly favored the genera *Arcobacter* and *Pseudomonas*.

The presence of SRB was also detected. According to *Sokolova et al. (2020)* the constant reinjection of PW results in contamination with sulfidogens, which can promote equipment corrosion and cause oil biodegradation.

In PW, we found that there were two genera common to all petroliferous regions (*Desulfotomaculum* and *Thermovirga*) (Fig. 4C). These genera are thermophilic and commonly found in oil reservoirs. The genus *Desulfotomaculum* comprises SRB that have been implicated in MIC processes (*Liu et al., 2019*) and are members of the class Clostridia of the phylum Firmicutes (*Whitman, 2009*). This genus has also been shown to use aromatic hydrocarbons (*e.g.*, toluene, *m*-xylene, *o*-xylene) as carbon and energy sources (*Morasch et al., 2004*). *Thermovirga* is a genus of the class Synergistia of the phylum Synergistetes, with a fermentative metabolism, using proteins, some amino acids, organic acids and alcohols as carbon and energy sources, and cystine and elemental sulfur as terminal electron acceptors (*Dahle & Birkeland, 2006*). These genera may be native of oil reservoirs, as they are common to all described petroliferous regions.

The greater distance of the R1 region when compared to the other regions indicates its unique features (Fig. 4A). This region exhibited the largest number of relevant microorganisms for the ordination, followed by R5, R6, and R7. The physicochemical characteristics, especially sulfate concentration, strongly influenced the microbial structure of these regions. The regions R1, R5 and R7 showed lower sulfate levels, while the R6 region showed higher sulfate levels (Supplementary Material 8).

R1 and R5 regions showed similar patterns: despite the low content of sulfur compounds, there was a dominance of genera reported in the literature as corrosion enhancers, such as SRB and $S^0$RB. There was also a high abundance of syntrophic and methanogenic archaea. Low concentrations or absence of sulfur compounds in PW may possibly decrease sulfate generation but do not prevent the development of

microorganisms that are involved in the carbon cycle (such as fermenting or syntrophic bacteria) (*Sokolova et al., 2020*).

There were positive correlations between SRB and MET. Some known SRB can syntrophically degrade organic substances such as lactate, ethanol, and propionate, in close association with hydrogenotrophic methanogens by means of $H_2$ (electron) transfer between species (*Plugge, Balk & Stams, 2002*), in the absence of sulfate. There may also be a syntrophic relationship between APB with methanogens and sulfidogens, since they consume the organic acids produced by APB. Correlations between genera in the same functional group may suggest that there are no competitive relationships. On the other hand, negative correlations may indicate that environmental characteristics can cause the selection of organisms that are both more and lesser adapted to the environment.

The R7 region also showed low sulfate content. Nonetheless, it was dominated by MET, APB and facultative hydrocarbon-degraders, with a significant number of mesophilic organisms. According to *Okoro, Samuel & Lin (2017)* methanogens dominate pipeline corrosion in low sulfate environments: some studies show that, in the absence of sulfate or nitrate, the fermentation of soluble organic compounds and oils into methane and carbon dioxide becomes the dominant metabolic processes (*Youssef, Elshahed & Mcinerney, 2009*).

Conversely, the R6 region had a low abundance of MET and this may be associated with the high relative abundance of SRB, since these organisms use and compete for the same substrates, hydrogen and acetate (*de Sousa Pires et al., 2021*; *Shelton et al., 2016*). The most relevant genus in this region was *Desulfovibrio*, a slow-growing aerotolerant SRB (*Crispim et al., 2018*), which are described as incomplete oxidants and responsible for souring and MIC (*Bedoya et al., 2021*; *Crispim et al., 2018*; *Semenova et al., 2019*). In addition, *Desulfuvibrio* species have already been detected in the PW samples from oil wells in China (*Menon & Voordouw, 2018*).

Positive correlations show that there is no competition between sulfidogenic groups and APB. In contrast, negative correlations indicate an inhibitory relationship between the SRB and NRB functional groups. This pattern is expected, since the injection of water treated with nitrate into oil reservoirs favors the development of NRB and, although the control mechanism is unclear, many proposed mechanisms include the competition between NRB and SRB for electron donors and carbon sources, inhibition of SRB by nitrite, changes in environmental redox potential, or even switching from conventional SRB metabolism to nitrate reduction (*Johnson, 2015*).

In the OW samples, R2 and R3 regions are located far from each other and from the other regions, which may indicate that they have unique characteristics (Fig. 5A). In addition, they also presented the largest vector, with the largest number of relevant genera. R2, R5, and R7 regions showed similar microbial patterns, with dominance of sulfidogens (SRB) and APB. In the R5 region there was also an expressive presence of IRB, and NRB were present in the R5 and R7 regions. This profile may have been defined by the physicochemical characteristics, mainly by the low sulfate levels.

*Nasser et al. (2021)* carried out a study in Saudi Arabia evaluating 11 samples, three of them of OW. The authors found that the composition of the microbial communities was
not determined by the facilities, but by the physicochemical conditions, specifically the sulfate concentration, since the reservoirs were rich in sulfate. Nevertheless, they also observed that other factors also influenced microbial dynamics. The presence of SRB in abundance in sulfate-poor environments was detected and could be explained by the likely syntrophy between these groups and methanogens. Furthermore, SRB can obtain energy using other metabolic pathways, such as fermentation and anaerobic respiration with alternative terminal electron acceptors. Otherwise, the scarcity of SRB in environments with high sulfate contents can be explained by the fact that when water is injected into wellhead, a mesophilic and possibly aerobic environment is created (*Liu et al., 2019*), or even by possible leaks in the facilities, which causes the presence of $O_2$ and can suppress the development of SRB (*Vigneron et al., 2016*).

*Summer et al. (2014)* evaluated four samples of OW originating from a rapidly corroding piping system and stored in tanks for a long period in the Gulf of Mexico. The tanks exhibited mesophilic temperatures, circumneutral pH, and low sulfate content. They observed dominance of genera of the IRB functional group and significant presence of the APB group, and both groups can participate in MIC processes. Nonetheless, the most dominant species was *Halanaerobium congolense* from the TRB group. Therefore, the authors concluded that the dominance of this functional group in rapidly corroding piping systems is an interesting finding, and it is necessary to consider the role of non-SRB sulfidogens in oil field corrosion.

*Duncan et al. (2017)* analyzed the microbial communities of OW samples at different points of the oil processing facility in the Gulf of Guinea and pointed out a similar microbial profile at all sampled points. The most abundant microorganisms are described as potential corrosion inducers. According to the authors, the low incidence of SRB suggests that souring in oil fields may be originated from other sulfidogens besides SRB. They concluded that monitoring at different points of oil facilities is an efficient way to identify corrosive organisms in problematic locations and thus to guide operational practices in order to control MIC.

In turn, the R3 region was dominated by methanogens, probably due to the application of chemical products, such as alkali-surfactant-polyacrylamide (ASP). *Gao et al. (2019)* performed a microbial profile comparison in reservoirs flooded with ASP and untreated IW, and reported a reduction in microbial diversity in ASP-flooded wells. This pattern was due to pH values above 11.5, representing extremely alkaline environments, which exceeds the survival limits of most microbial species. Therefore, that specific and highly stressful environment promoted the selection of alkali-tolerant organisms, such as NRB *Halomonas*, MET *Methanobacterium* and properly alkaliphilic genera that can use sulfur compounds as electron acceptors, such as *Anoxynatronum* (*Toh et al., 2011*). Regarding the wells not flooded with ASP, a dominance of genera commonly found in oil reservoirs flooded with water was observed. They concluded that the microbial profile of reservoirs treated with ASP was determined by the high pH of the environment, causing a limitation of microbial growth to a certain extent, with a selection of organisms resistant to high pH values. Hence, these ASP-flooded reservoirs would, at first, be unfeasible for nitrate

application to favor microorganisms which can compete with those who have a corrosive metabolism.

In O samples, the distance between the regions indicates that each one presented a distinct microbial profile. Furthermore, the opposition between R3 and R6 suggests that they may be completely negatively correlated, and there are no shared genera between these regions. The R3 region was dominated by aerobic microorganisms, with the vast majority not known to directly participate in MIC, whereas the R6 region was dominated by microorganisms that are known to cause MIC (sulfidogens).

*Liu et al. (2019)* evaluated two samples in replicates of a virgin field (without IW nor recirculation of PW) and one at the wellhead of a well with continuous water injection, in an oil field located in Northeast China. The authors reported a dominance of Proteobacteria in both samples. In the virgin well, there was a higher abundance of the genera *Ochrobactrum* and *Acinetobacter*, while at the wellhead of the other well, there was a higher abundance of genera such as *Lactococcus* and *Pseudomonas*. Among these genera, *Pseudomonas* is the most mentioned in oil fields, probably due to the water injection into the reservoirs, and by the fact its species are potential hydrocarbon degraders. Strictly aerobic species of the genus *Sediminibacterium* were isolated from a Chinese reservoir and display interesting characteristics, with species capable of producing hydrogen sulfide ($H_2S$) and organic acids from sugars. Regarding the other genera, there is not much information related to their metabolism in the oil field; however, some species of these genera are aerobic or facultative anaerobes, antibiotic-resistant and are also considered pathogens, as well as important organisms in biotechnology.

In contrast, the R6 region was dominated by SRB *Archaeoglobus* and *Desulfovibrio* and showed a high abundance of APB and $S^0$RB, indicating that there may be a syntrophic relationship between these functional groups. *dos Santos et al. (2020)* evaluated the microbial profile of an oil sample from a Brazilian field and also analyzed the metabolic profile of these samples. The authors pointed out the dominance of sulfate-reducing *Archaeoglobus* and *Desulfovibrio*. The genus *Archaeoglobus* comprises hyperthermophilic sulphate-reducing archaea, frequently found in hydrothermal vents and oil wells, with temperatures around 85 °C, which develop in environments with high salinity (*Birkeland et al., 2017*). And, despite the authors having found dominance of SRB, the dominant metabolic pathway was sulfite metabolism.

R7 region had the lowest number of genera that explain data variability data (Fig. 6A). The high uniqueness of the genera (Fig. 6B) can be explained by data heterogeneity, representing, only in this region, 60% of all studies. That is, in addition to the intrinsic differences in the physicochemical conditions existing between the samples from the same oil field (Supplementary Material 8), there were also differences between oil fields, which displayed very different microbial profiles. This region exhibited dominance of $S^0$RB, APB, NRB, and hydrocarbon degrading genera. The high availability of nutrients, together with the probable decrease in pH due to the excess of byproducts of the metabolism of both APB (organic acids) and $S^0$RB species ($H_2S$), may have excluded SRB and MET. Moreover, sulfate unavailability may have been an important factor in the low abundance of SRB (Supplementary Material 8).

In the SD samples, the presence of the genus *Marinobacter* in all regions may indicate that this genus actively participates in the biochemical processes that occur in oil fields. The presence of hydrocarbon-degrading bacteria and biofilm-forming syntrophic bacteria allied to MET is expected to facilitate hydrocarbon degradation in oilfields, since it is likely that, while syntrophic bacteria degrade hydrocarbon substrates to products such as acetate, hydrogen or carbon dioxide, methanogens use these substrates to produce methane (*Fowler, Toth & Gieg, 2016*).

The R2 region was contemplated by only one study (*Albokari et al., 2014*), including only one sample from an oil field of the Eastern Province of Saudi Arabia. The correlations between genera of versatile, hydrocarbon-degrading, nitrate-reducing, and sulfidogenic bacteria (*Chromohalobacter*) with genera from the TRB group (*Halanaerobium*) may indicate that there is no competition between these genera and that the environment is providing necessary conditions for the development of both genera.

In contrast, the negative correlation between the genus *Halomonas*, with facultative aerobes, nitrate-reducing, halophilic and alkane-degrading species, and the genus *Methanosarcina*, a methanogen that can use three metabolic pathways (acetoclastic, hydrogenotrophic and methylotrophic) to produce methane (influential in the R6 region) may indicate inhibition by toxicity. Nitrite is an intermediary compound in denitrification by NRB, and (*Detlef & Conrad, 1998*) evaluated the effect of nitrite application on methanogenesis in pure cultures of *Methanosarcina*, demonstrating that this compound inhibits methanogenesis.

*Albokari et al. (2014)* reported dominance of a non-culturable genus, *Flavobacterium* (phylum Bacteroidetes), with a relative abundance of 66%, and this phylum has chemolithotrophic and anaerobic representatives. Furthermore, the authors only reported their results at the phylum level, and, thus, it was not included in our initial data analysis (see Materials and methods section). The abundance of *Flavobacterium* may be due to the high availability of organic material rich in dimethyl sulfide (DMS) originated from the degradation of the compatible solute dimethylsulfonyiopropionate (DMSP). DMS can be co-metabolized, generating a flow of electrons that is probably used in energy metabolism and thus in biomass formation (*Green et al., 2011*).

The R3, R4, R5, R6 and R7 regions had in common the genera *Marinobacter*, *Pseudomonas* and *Acetobacterium*. In these regions, the high abundance of *Pseudomonas* suggests that these organisms triggered the formation of biofilms, which harbor diverse microorganisms with metabolic capability of promoting or potentiating MIC processes (*Hamzah et al., 2014*). *Salgar-Chaparro et al. (2020),* who analyzed metabolism products in corroded gaskets on an offshore platform, observed a dominance of *Pseudomonas* and *Shewanella*. *Pseudomonas* species are pioneering colonizers in biofilm formation and are capable of synthesizing extracellular polymeric substances (EPS), such as adhesive exopolysaccharides, and these compounds promote the adherence of other microbial groups, establishing a cooperative relationship, in addition to promoting protection against biocides, disinfectants, and other environmental stresses (*Alena, Schreiberova & Masa, 2014*). In these biofilms, metal cations are bound by the EPS, which increases ionization and ion accumulation, and, thus, altering the properties of metals and

promoting corrosion. Additionally, the production of electron mediators may favor the transfer of electrons between cells and metals and stimulate corrosion rates by the extracellular electron transfer MIC mechanism (EET-MIC). This mechanism has been recently discovered and is an electrical biocorrosion performed by chemolithotrophic microorganisms that use metallic iron (Fe0) as an electron donor, oxidizing it to Fe2+ (*Kato, 2016*; *Kato, Yumoto & Kamagata, 2014*). This electron extraction can occur through direct contact with the metallic surface—through cytochrome multi-heme enzymes linked to the cell membrane or to nanowires—or indirectly—through the diffusion of H2 or soluble redox carriers (such as flavins, phenanzines and quinones) to the extracellular medium (*Kato, 2016*). Nevertheless, *Pseudomonas* bears a metabolism capable of promoting corrosion under anaerobic conditions, such as iron oxidation combined with nitrate reduction or nitrite production. Nitrate reduction could explain the low abundance of SRB in this region since SRB are sensitive to the presence of nitrite (*Johnson, 2015*; *Salgar-Chaparro et al., 2020*).

R4 and R6 regions showed an abundance of thermophilic $S^0$RB and SRB. In both regions there were statistically positive correlations between methanogens, BPA and BNR. In the R4 region, the positive correlations between methanogens and NRB, APB, $S^0$RB and IRB may indicate that local microbiota are engaged in syntrophies. Nonetheless, despite the methanogens being abundant in these sample types, when specifically analyzing the documents, it is possible to observe that in this region they were not so relevant as the sulfidogens and hydrocarbon degraders.

*Bonifay et al. (2017)* performed a metagenomic and metabolomic analysis on pig solid deposits at different depths in two adjacent pipeline systems with different corrosion rates (high and low) in a North Sea oil field. Both systems were similar in terms of physicochemical characteristics, which suggests that abiotic processes may not be the differentiating factor, but that differences in corrosion rates can be explained by the biological activity that occurs in both systems. The authors used the electron transport chain to explain these differences, because they observed a dominance of genera capable of reducing nitrate in pipelines with low corrosion rates and dominance of sulfidogens in pipelines with high corrosion rates. Although this study clarified the metabolic activity of organisms in pipelines with different corrosion rates, the process that triggered the dominance of different organisms in apparently similar environments remained unclear. In the R6 region, the negative correlation between *Methanosarcina* and *Halomonas* (more abundant in the R2 region) may indicate exclusion by toxic metabolites. As previously discussed, in the R2 region, NRB may release nitrite, which is toxic to methanogens, such as *Methanosarcina*, causing their inhibition (*Detlef & Conrad, 1998*).

The R7 region displayed the largest vector (Fig. 7A). It was represented by 11 articles, all of them from Nigeria, with characterization of 21 SD samples. This region was the one with the highest number of physicochemical information (Supplementary Material 8). This region was dominated by hydrogenotrophic methanogens (*Methanobacterium*, *Methanocalculus*, *Methanoculleus*, and *Methanosaeta*), commonly found in environments of low salinities and mesophilic temperatures. There was also the presence of methylotrophs, such as *Methanolobus*, which use methanol and methylamine to produce

methane, which are not used by SRB (*Duffes, Jenoe & Boyaval, 2000*; *Jeon, Yi & Park, 2014*). Therefore, the high abundance of hydrocarbon degraders and methanogens, and their positive correlation, may indicate a syntrophic relationship between nutrient availability by hydrocarbon degraders and consumption by methanogens.

Several factors can influence methanogenesis, such as salinity, temperature, pH value, substrate availability, microbial interactions, and anthropic interventions (*Blake et al., 2015*). *Okoro, Samuel & Lin (2017)* evaluated the effects of temperature and pH on corrosion rates and methane production using metallic coupons. The authors reported that methane production strongly correlated with corrosion rates and moderately with pH. The rise in temperature can cause a shift from acetoclastic methanogenesis to hydrogenotrophic methanogenesis (*Fey & Conrad, 2003*). Although temperatures above 55 °C inhibit methanogenesis, it is still unclear how temperature influences methane production pathways in different ecosystems (*Okoro, Samuel & Lin, 2017*). High abundances of methanogens are uncommon in environments with high concentrations of saline sulfate (*Gittel et al., 2009*), since SRB can develop and compete with MET for electron donors and, therefore, prevailing sulfate scarcity, methanogens thrive. In Nigeria's oil fields, THPS is commonly used as biocide (*Okoro, Samuel & Lin, 2016*), and a direct relationship between THPS use and the predominance of methanogens is highly probable. At least three hypotheses capable of explaining this phenomenon are described.

First, limitations regarding the effectiveness of THPS against methanogens were detected. The THPS dosage allows the elimination of part of the microbial population, but some survive and recover, and may develop resistance. Resistance of some groups of methanogens to this compound has already been verified, such as the genus *Methanolobus* (*Okoro, Nwezza & Lin, 2018*). Second, continuous nitrate and THPS dosage may have suppressed the SRB population due to exclusion by competition for NRB groups, even in sulfate-rich environments, or by sensitivity to THPS. And third, THPS can serve as a precursor in methanol production and, thus, enhance the proliferation of methanogens (*Machuca & Salgar-Chaparro, 2019*).

*Machuca & Salgar-Chaparro (2019)* conducted a study to identify the diversity and metabolic capabilities of the microbiota of solid deposits in a pipeline in an Australian oil field. They used THPS and acrolein as biocides and reported dominance of methanogens and non-SRB sulfidogens. Additionally, they pointed out the presence of genes referring to the energy metabolism of methane, that is, a high capacity of these microorganisms to use methane pathways for oil degradation. The authors explained that methanol is the most likely byproduct of THPS degradation, and this compound may be a potential carbon source for methanogens. Among the four samples evaluated by the authors, three described the dominance of methanogens and non-SRB sulfidogens. High sulfate content was reported only in one, which may have created a competitive relationship between the groups, since there was an overlap of the SRB in relation to the methanogens; both with high abundances of S⁰RB, since all samples exhibited high levels of other forms of sulfur. There were pathways related to primary metabolism (Biosynthesis of Carbohydrates, Amino Acids, Nucleotides, Fatty acids, Cofactor, Transporters and Vitamins and Cellular

structure, Glycolysis, Fermentation, Electron transport chain, among others) and secondary (Biosynthesis of Terpenes, Precursors, Tetrapyrrole, among others) (Fig. 8).

Primary metabolism pathways are responsible for maintaining bacterial survival (*Pereira, Pilz-Junior & Corção, 2021*). Cofactor, Carrier and Vitamin Biosynthesis, Nucleoside and Nucleotide Biosynthesis, Carbohydrate Biosynthesis, Amino Acid Biosynthesis and Fatty acid and Lipid biosynthesis were dominant in the oil fields (Fig. 8). Among them, only the Cofactor, Carrier and Vitamin Biosynthesis refer to secondary metabolism (*Pereira, Pilz-Junior & Corção, 2021*). These are produced for survival under certain conditions, such as microbial competition for nutrients and/or space, and protection from abiotic stress (*Pereira, Pilz-Junior & Corção, 2021*; *Singh et al., 2019*). Nevertheless, the products generated from this pathway are related to the enzymatic reactions of amino acid and carbohydrate synthesis (*Alves et al., 2013*; *Santos et al., 2018*), which are also related to life maintenance (primary metabolism). The biologically active variants of vitamin $B_{12}$ contain rare organometallic bonds, which are used by enzymes in a variety of central metabolic pathways, such as amino acid (L-methionine) synthesis (*Santos et al., 2018*). In turn, vitamin $B_1$ is important in carbohydrate metabolism, besides being an essential growth factor for some bacteria and archaea (*Alves et al., 2013*).

In contrast, pathways related to nucleotide and carbohydrate metabolism are directly involved in the hydrolysis and initial breakdown of complex compounds and substrates (*Yang, 2020*). Amino acids can be used as a nitrogen source by microorganisms (*Xia et al., 2021*). Moreover, some synthesis pathways of sulfur-containing amino acids (L-methionine and L-lysine) were found, which participate in sulfur metabolism; many organisms use this pathway, such as sulfidogens (*Pereira, Pilz-Junior & Corção, 2021*), which may explain the high activity of amino acid biosynthesis in oil fields. Finally, the synthesis of some fatty acids is critical for the maintenance of membrane structure and function in many bacterial groups (*Yang, 2020*), and this metabolic profile is supported by the findings of *Roy et al. (2018)*.

*Roy et al. (2018)* evaluated the metabolic profile of hydrocarbon-rich oil refinery sludge in Northeast India and found a community similar to the present study, with an abundance of anaerobic and fermentative organisms. The authors highlighted microbial populations of sulfate reducers, syntrophic microbes, and methanogens. They also observed high activity of metabolism of carbohydrates, amino acids, energy, cofactors and vitamins, nucleotides, lipids, biodegradation of xenobiotics, terpenoid metabolism, metabolism and biosynthesis of secondary metabolites.

The pairwise combination of sample types from the oil fields was also evaluated to biologically verify the statistically significant differences between the metabolic groups. The high number of statistically different pathways in the combination between PW and SD can be explained by the greater number of articles and samples obtained, which led to a greater number of 16S rRNA sequences for these sample types. Carbohydrate, amino acid, and nucleotide metabolism are associated with hydrolysis, the initial breakdown of complex compounds and substrates (*Ijoma et al., 2021*). In the PW and SD samples, the intense activity of these pathways may be associated with the existing syntrophic relationships between APB, sulfidogens, and methanogens. Initially, the complex

compounds are hydrolyzed by fermentative organisms (such as APB). This group uses only the glycolytic pathway to metabolize organic compounds. Hence, they release organic acids (acidogenesis), acetate and $H_2$ (acetogenesis) as end-products of their metabolism, which are used as substrates by the chemolithotrophic SRB and methanogenic bacteria and archaea, through the incomplete reductive tricarboxylic acid cycle (TCA) cycle pathway.

Incomplete reductive pathways of the TCA cycle can still transform pyruvate into biosynthetic intermediates needed in response to anaerobic or microaerophilic growth conditions (*Wood, Aurikko & Kelly, 2004*). Most anaerobic autotrophic bacteria and archaea do not have the ability to carry out a complete oxidative or reductive TCA cycle (*Goodchild et al., 2004*; *Wood, Aurikko & Kelly, 2004*). This pathway was grouped in class 2 as C1 Compound Utilization and Assimilation. This pathway is also statistically significant between IW and PW samples. Nonetheless, only between PW and SD, the incomplete reductive TCA cycle occurred (Fig. 8). This does not mean that this pathway is not active in other substrate types. It is necessary to consider the limitations of the current study, such as the low number of IW, OW and O samples.

Electron transfer chains/aerobic metabolic pathway was significant among the combinations of IW/O, SD/O, PW/O, and PW/SD samples. This pathway is essential for aerobic respiration in microorganisms, and it is the pathway that provides the highest energy yield. In oil fields, high activity of this pathway is not very common, since they are environments deficient in $O_2$. In IW samples, the statistical significance of this pathway was already expected due to the origin and processes of this sample type in oil fields, due to water injection in wellhead, creating an aerobic environment (*Liu et al. 2019*) or by pits along pipelines (*Vigneron et al., 2016*).

In turn, there was a low abundance of this pathway (Electron transfer chains/aerobic) in PW when compared to IW. Nevertheless, the presence of this metabolic pathway in both sample types was probably due to the recirculation of PW together with IW, which can promote contamination of organisms such as *Pseudomonas*, which are versatile and adapt to anaerobic environments (*Bedoya et al., 2021*).

In contrast, this metabolic pathway is less abundant in O samples and SD. Nevertheless, the explanation for their occurrence may be due to the pits along the pipelines, which can cause an aerobic microenvironment and permit the development of organisms with this kind of metabolism (*Vigneron et al., 2016*). Another possible explanation would be the sampling location, as explained by *Liu et al. (2019)*, who found a high amount of *Lactococcus* and *Pseudomonas* at the wellhead of an oil well with continuous injection of IW.

## CONCLUSION

The microbial composition of oil fields in different sample types and petroliferous regions, along with the production and transport processes worldwide, showed that most of detected microorganisms are potentially involved in MIC processes (sulfidogenic, biofilm forming, acidogenic, acetogenic, and methanogenic microbes). Furthermore, these microbial functional groups, jointly with nitrate reducers, probably established syntrophic or inhibitory relationships.

The PW and SD samples had the largest data volume and, thus, more consistent results were obtained from these samples. In PW samples, despite high microbial diversity, sulfidogenic groups were more abundant, with the presence of the *Desulfotomaculum* and *Thermovirga* genera in all petroliferous regions. Moreover, there was an emphasis on the syntrophic relationships between sulfidogens, methanogens, nitrate-reducing and acid-producing microorganisms. Otherwise, in SD samples, despite also presenting high diversity, a distinct microbial profile was observed: higher abundance of APB and NRB, and important syntrophic associations between sulfidogens, methanogens and NRB were retrieved. The latter group includes biofilm-forming organisms, such as *Pseudomonas*. In SD samples, the NRB *Maribacterium* was present in all petroliferous regions. Presence and abundance of these groups and correlations between them were influenced by environmental physicochemical characteristics, especially, sulfate concentration, as well as anthropic interferences, such as the application of biocides. High sulfate content favored the sulfate reduction metabolism by SRB, while in sample types with low sulfate content, there was a metabolic deviation of SRB, which probably ally with methanogens as a survival strategy. The application of chemical products caused several changes in the environment. In many cases this strategy is efficient in causing the inhibition of potential organisms involved in MIC (such as SRB), but it induced the selection of: (i) antibiotic resistant organisms, such as *Pseudomonas*, (ii) organisms whose metabolism uses the applied products (THPS) as substrate, such as *Methanolobus*, and (iii) organisms tolerant to extreme conditions, mainly high pH values, such as *Halomonas*, *Methanobacterium*, and *Anoxynatronum*, among others. Moreover, the petroleum environment was also rich in non-SRB sulfidogens, and these bacteria are probably not affected by common biocides. According to the functional prediction analysis, there are high microbial activity in oil fields, with the highest abundances of Cofactor, and Carrier, and Vitamin Biosynthesis, which plays the role of catalysts for essential reactions in microorganisms, such as the biosynthesis of carbohydrates and amino acids. Additionally, there was also the presence of secondary metabolic pathways, such as terpenoid biosynthesis and aromatic compounds biosynthesis, which are directly involved in survival, under stressful conditions as well as used as a defense mechanism.

Therefore, our worldwide systematic review and analyses of existing data in the specialized literature of microbial communities associated with environments of the oil and gas industry has generated a global scientific synthesis of this theme, which can greatly facilitate the establishment of standardized approaches to control microbial-induced corrosion.

## ACKNOWLEDGEMENTS

We would like to thank all the people that contributed directly or indirectly to this study.

### Funding

This work was supported by PETROBRAS (SIGITEC 2017/00587-2; ANP n° 21035-1), the Graduate Programs of Microbiology and Bioinformatics, and the Pro-Rectory of Research of the Universidade Federal de Minas Gerais (UFMG). The funders had no role in study design, data collection and analysis, decision to publish, or preparation of the manuscript.

### Grant Disclosures

The following grant information was disclosed by the authors:
PETROBRAS: SIGITEC 2017/00587-2; ANP n° 21035-1.
Graduate Programs of Microbiology and Bioinformatics.
Pro-Rectory of Research of the Universidade Federal de Minas Gerais (UFMG).

### Competing Interests

Vinicius Waldow, Claudia Groposo, Rubens N. Akamine, and Maira Sousa are employees of PETROBRAS.

### Author Contributions

- Joyce Dutra conceived and designed the experiments, performed the experiments, analyzed the data, prepared figures and/or tables, authored or reviewed drafts of the article, and approved the final draft.
- Rosimeire Gomes conceived and designed the experiments, performed the experiments, analyzed the data, prepared figures and/or tables, authored or reviewed drafts of the article, and approved the final draft.
- Glen Jasper Yupanqui García conceived and designed the experiments, performed the experiments, analyzed the data, prepared figures and/or tables, authored or reviewed drafts of the article, and approved the final draft.
- Danitza Xiomara Romero-Cale conceived and designed the experiments, performed the experiments, analyzed the data, prepared figures and/or tables, authored or reviewed drafts of the article, and approved the final draft.
- Mariana Santos Cardoso conceived and designed the experiments, performed the experiments, analyzed the data, authored or reviewed drafts of the article, and approved the final draft.
- Vinicius Waldow conceived and designed the experiments, analyzed the data, authored or reviewed drafts of the article, and approved the final draft.
- Claudia Groposo conceived and designed the experiments, analyzed the data, authored or reviewed drafts of the article, and approved the final draft.
- Rubens N. Akamine conceived and designed the experiments, analyzed the data, authored or reviewed drafts of the article, and approved the final draft.
- Maira Sousa conceived and designed the experiments, analyzed the data, authored or reviewed drafts of the article, and approved the final draft.

- Henrique Figueiredo conceived and designed the experiments, analyzed the data, authored or reviewed drafts of the article, and approved the final draft.
- Vasco Azevedo conceived and designed the experiments, analyzed the data, authored or reviewed drafts of the article, and approved the final draft.
- Aristóteles Góes-Neto conceived and designed the experiments, analyzed the data, authored or reviewed drafts of the article, and approved the final draft.

## Data Availability

The raw data are available in Supplemental Files.

## Supplemental Information

Supplemental information for this article can be found online at http://dx.doi.org/10.7717/peerj.14642#supplemental-information.

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
