# Peer review of "Corrosion-influencing microorganisms in petroliferous regions on a global scale: systematic review, analysis, and scientific synthesis of 16S amplicon metagenomic studies"

_PeerJ, doi:10.7717/peerj.14642_

## Round 0.1 · original submission · Major Revisions

Dear Dr. Dutra and colleagues:

Thanks for submitting your manuscript to PeerJ. I have now received two independent reviews of your work, and as you will see, the reviewers raised some concerns about the research. Despite this, these reviewers are optimistic about your work and the potential impact it will have on research studying the systematics of corrosion-influencing microbes. Thus, I encourage you to revise your manuscript, accordingly, taking into account all of the concerns raised by both reviewers.

There are many comments by both reviewers that ask for more information on specific issues; please address these. This especially applies to the statistical analyses, selection of certain databases over others, and provision of genes within metabolic pathways.

Please ensure all figures and tables are clearly resolved.

I look forward to seeing your revision, and thanks again for submitting your work to PeerJ.

Good luck with your revision,

-joe

Reviewer 1 ·

Basic reporting

This is an interesting and well-written article. The background and motivation of the study were stated clearly. The systematic literature review conducted in this study was well-designed and followed the standard procedures of systematic literature view. The databases and search criteria used in the literature review were appropriate.

The authors also conducted quantitative analysis using a couple of statistical analysis, including correlation analysis, principal component analysis and prediction model. The statistical methods were used appropriately and technically correct. The results of these statistical analyses were explained clearly, and the interpretation of the results were sound. And I appreciate the authors made use of visualization tools to make the results more presentable and interpretable.

One specific question in Table 2. For the column – Articles (n), why was 2 shown as “two”? Was this intended?

On PCA, the first two PCs only explained ~40-50% of data variability in the data. Are the first two PCs sufficient to explain the variability in the data? Were there any of important components or aspects not captured by the analysis? What’s the percentage of variation explained by the 3rd and 4th PC in the analysis?

Experimental design

See comments above

Validity of the findings

See comments above

Reviewer 2 ·

Basic reporting

Your systematic work of the summarized microbiomes of the oil-producing areas is excellent, but there are a few problems.

1. There have been various tools for studying amplicon function prediction, such as PICRUSt, PICRUSt2, Tax4Fun, etc. Why using PICRUSt2 as a tool for your functional prediction research? Please explain in the introduction.
2. In the literature research section, you used the search method to collect 540 documents, borrowed programming tools in the screening section, and selected 69 papers for subsequent research and analysis. These results are based on the three databases (Scopus, Web of Science, and OnePetro databases) you selected. The PubMed database contains many studies in the field of microbiology. Therefore, in the literature collection phase, whether the database was investigated.
3. Environmental changes can cause changes in microbial community structure. About the changes it caused, this study also mentioned. Studies have repeatedly said that different concentrations of sulfate and nitrate can cause changes in community structure. However, the designations for various concentration levels are not listed.
4. The article discusses metabolic pathways associated with corrosion but lacks discussion of functional genes. In microbial corrosion, microorganisms must first adhere to the metal surface. Please explain whether there are significant active genes in your survey, such as those related to biological adhesion and material transport.
5 The EET-based MIC mechanism is an emergering perspective to explain how MIC occured, please refer to EET, to make sufficient discussion which may help to improve the quality of the manuscript.



There are some problems with article chart readability and quantifiers.
1. Some data forms are not standardized
2. Figure: The text of the picture in the article is not clear enough for reading
3. Table 1: In the table, there is a problem with the font usage specification
4. Table 2: The region column has Arabic numbers and text coexistence, such as “two” or “2.”

Experimental design

No comments

Validity of the findings

No comments

---

## Round 0.2 · accepted · Accept

Dear Dr. Dutra and colleagues:

Thanks for revising your manuscript based on the concerns raised by the reviewers. I now believe that your manuscript is suitable for publication. Congratulations! I look forward to seeing this work in print, and I anticipate it being an important resource for groups studying the systematics of corrosion-influencing microbes. Thanks again for choosing PeerJ to publish such important work.

Best,

-joe